# The gut microbiome-prostate cancer crosstalk is modulated by dietary polyunsaturated long-chain fatty acids

Gabriel Lachance[1,2,3], Karine Robitaille[1,2], Jalal Laaraj ®[1,2], Nikunj Gevariya[1,2], Thibault V. Varin[3], Andrei Feldiorean[4,5], Fanny Gaignier[1,2], Isabelle Bourdeau Julien[6], Hui Wen Xu[7], Tarek Hallal[4,8], Jean-François Pelletier[1,2], Sidki Bouslama[9], Nadia Boufaied[4], Nicolas Derome[9,10], Alain Bergeron ®[1,2], Leigh Ellis[11], Ciriaco A. Piccirillo[12,13], Frédéric Raymond ®[6], Yves Fradet[1,2], David P. Labbé ®[4,5,8], André Marette ®[3] & Vincent Fradet ®[1,2,6] ✉

The gut microbiota modulates response to hormonal treatments in prostate cancer (PCa) patients, but whether it influences PCa progression remains unknown. Here, we show a reduction in fecal microbiota alpha-diversity correlating with increase tumour burden in two distinct groups of hormonotherapy naïve PCa patients and three murine PCa models. Fecal microbiota transplantation (FMT) from patients with high PCa volume is sufficient to stimulate the growth of mouse PCa revealing the existence of a gut microbiome-cancer crosstalk. Analysis of gut microbial-related pathways in mice with aggressive PCa identifies three enzymes responsible for the metabolism of long-chain fatty acids (LCFA). Supplementation with LCFA omega-3 MAG-EPA is sufficient to reduce PCa growth in mice and cancer up-grading in pre-prostatectomy PCa patients correlating with a reduction of gut Ruminococcaceae in both and fecal butyrate levels in PCa patients. This suggests that the beneficial effect of omega-3 rich diet is mediated in part by modulating the crosstalk between gut microbes and their metabolites in men with PCa.

A limited number of correlative studies have reported differences in the gut microbiota of healthy versus prostate cancer (PCa) patients[1-3]. Interestingly, many of the tumour-derived bacteria sequences recently found in prostate cancer tissues are naturally present in human stool, suggesting that gut microbes could be directly or indirectly impacting tumour development. Furthermore, PCa treatments such as androgen deprivation therapies (ADT) are associated with gut microbiome imbalances[4-6]. Recent work showed that ADT could drive commensal bacteria, including species from the Ruminococcus genus to synthetize androgen from circulating precursors to fuel castration-resistant

[1]Laboratoire d'Uro-Oncologie Expérimentale, Oncology Axis, Centre de recherche du CHU de Québec-Université Laval, Québec, QC, Canada. [2]Centre de recherche sur le Cancer de l'Université Laval, Québec, QC, Canada. [3]Centre de recherche de l'IUCPQ, Québec, QC, Canada. [4]Cancer Research Program, Research Institute of the McGill University Health Centre, Montréal, QC, Canada. [5]Division of Urology, Department of Surgery, McGill University, Montréal, QC, Canada. [6]Institute of nutrition and functional foods (INAF) and NUTRISS Center - Nutrition, health and society of Université Laval, Québec, QC, Canada. [7]Department of Mathematics and Statistics, Université Laval, Québec, QC, Canada. [8]Department of Anatomy and Cell Biology, McGill University, Montréal, QC, Canada. [9]Institut de Biologie Intégrative et des Systèmes (IBIS), Université Laval, Québec, QC, Canada. [10]Department of Biology, Université Laval, Québec, QC, Canada. [11]Center for Prostate Disease Research, Murtha Cancer Center Research Program, Department of Surgery, Uniformed Services University of the Health Sciences and the Walter Reed National Military Medical Center; The Henry M. Jackson Foundation for the Advancement of Military Medicine, Inc., Bethesda, MD, USA. [12]Infectious Diseases and Immunity in Global Health Program, Research Institute of the McGill University Health Centre, Montréal, QC, Canada. [13]Department of Microbiology and Immunology, McGill University, Montréal, QC, Canada. ✉e-mail: Vincent.Fradet@crchudequebec.ulaval.ca

prostate cancer (CRPC) development[7]. Treatment of CRPC patients with abiraterone acetate, a second-generation anti-androgen drug, was shown to be associated with an enrichment of Akkermansia muciniphila in patient's gut microbiota[4]. Gut Akkermansia muciniphila was reported to have a beneficial impact on human health[8] and to improve response to novel immunotherapies in some cancers[9]. Altogether, these studies support that the gut microbiota plays a role in response to hormonal treatments in PCa patients. However, the potential interplay between gut microbiota and PCa development in patients without ADT remains understudied.

PCa is the fourth most diagnosed cancer worldwide and second most common cancer in men[10]. PCa progression is most often slow over many years even under surveillance and the potential benefit of curative treatments such as radical prostatectomy or radiotherapy is offset by potential long-term detrimental impacts on patient's quality of life. Likewise, cancer recurrence after radical prostatectomy or radiotherapy as detected by a rise of the prostate-specific antigen (PSA) blood levels also has a highly variable course, often lasting many years before clinically manifest metastases are detected. Consequently, there are important long-term benefits associated with the development of new primary or secondary prevention therapies to delay the need for primary treatments or ADT for systemic recurrences. Numerous studies provided support for the involvement of modifiable factors, such as the diet, in PCa progression. For example, a diet rich in animal fat containing higher amount of saturated fat was associated with increased risk of lethal PCa compared to a diet enriched in polyunsaturated fat from plant[11]. Fatty acid assimilation in humans is regulated by the commensal gut bacteria[12,13]. Hypercaloric diets, such as high-fat diet (HFD), stimulate the proliferation and activity of gut microbes, increase intestinal permeability, and are associated with bacteria-derived molecule leakage (endotoxemia) ultimately contributing to chronic inflammation[12–14]. In addition, several immunocompetent genetically engineered mouse models of PCa develop more aggressive tumours when fed an HFD enriched in animal/saturated fat[15–17]. On the contrary, we and others showed that omega-3 rich diet was sufficient to reduce PCa progression in mouse models[18–20], supporting the rationale for dietary interventions to hinder PCa development. How diet affects PCa growth and whether the gut microbiota is involved in the interaction remains, however, largely unknown.

Here, we found a reduced fecal microbiota alpha-diversity associated with increased PCa volume and aggressiveness in two distinct groups of ADT naïve PCa patients and three distinct syngeneic murine PCa models. Human fecal microbiota transplantation (FMT) from patients with PCa stimulated the growth of mouse PCa, supporting a microbiota-PCa crosstalk. The analysis of commonly altered fecal metabolic pathways in mouse models with high tumour volume identified three enzymes involved in the degradation of long-chain fatty acids (LCFA). We leveraged fecal samples provided by a subgroup of patients from a larger cohort of 130 men diagnosed with GG ≥ 2 prostate cancer and undergoing radical prostatectomy[21]. Participants from this phase II double-blind randomized placebo-controlled trial were randomized to receive MAG-EPA omega-3 or placebo prior to radical prostatectomy[21]. In parallel, we treated a syngenic mouse model with the same omega-3 supplement prior to prostate tumour cell injection. Dietary supplementation with LCFA omega-3 reduced tumour growth in a mouse model and reduced cancer upgrades at prostatectomy in PCa patients and was associated with a decrease in fecal levels of Ruminococcaceae in both. Analysis of metabolites produced by the Ruminococcaceae family of bacteria in patients' fecal samples treated with an omega-3 supplement identified a selective reduction of butyrate. On the other hand, fecal butyrate levels were increased in patients with early metastatic PCa. Butyrate is mainly produced by gut bacteria and may represent an important mediator of the gut microbiota-PCa crosstalk. It also

identifies gut butyrate as one mechanism by which omega-3 rich diet exerts anti-PCa activity.

## Results

### PCa tumour volume and aggressiveness are associated with alterations of the gut microbiota

**PCa patients.** To test if changes in gut microbiota were associated with PCa tumour volume and aggressiveness, we profiled the microbial content of fecal samples collected from PCa patients without ADT at different stages of the disease. The first population of 62 patients with a localized PCa (ISUP Grade Group (GG) ≥2) provided a fecal sample 5.0 weeks (±0.5) on average before radical prostatectomy (Supplementary Table 1). Based on the surgical pathology report, fecal samples were separated in tertiles of low (3.2 ± 1.4 g), medium (7.4 ± 1.4 g) or high (18.7 ± 12.7 g) PCa tumour volume (% of cancer area multiplied by weight (g) of the radical prostatectomy specimen) (Fig. 1a). 16S rRNA profiling was performed and microbiota profiles were grouped in accordance with the tumour volume at the radical prostatectomy for each patient (Fig. 1b). 16S rRNA metataxonomic analysis did not show differences in the major phylum such as Firmicutes and Bacteroidetes (Fig. 1b). To further evaluate shifts in the whole-microbiota population, we used Shannon alpha diversity reference index[22,23]. The Shannon diversity index was reduced by 8.3 ± 2.6% for patients in the highest tumour volume compared to patients in the low tumour volume group (Fig. 1c, $p = 0.02$, two-sided Welch's $t$-test). To test if patients with high tumour volume systematically displayed changes in their microbiota, we measured the beta diversity using the Bray-Curtis index for the whole cohort. Principal component analysis (PCA) did not show difference in beta diversity for those patients (Fig. 1d, $p = 0.48$, Permanova).

We next performed similar studies on fecal samples obtained from 47 PCa patients on average 5.2 ± 1.0 years post-prostatectomy (see Supplementary Table 2 for clinical and pathological characteristics at prostatectomy). At the time of fecal sample collection, 20 patients had undetectable PSA, *i.e.* no biochemical recurrence (BCR) for a minimum of 5 years post-surgery, 9 patients had very low PSA values (0.17 ± 0.02 ng/mL) and 18 patients had significantly high PSA values (2.96 ± 0.46 ng/mL) (Fig. 1e). Of note, patients with this high level of PSA after prostatectomy have systemic metastasis when tested with the most recent nuclear medicine procedures such as Prostate Specific Membrane Antigen Positron Emission Tomography (TEP-PSMA) imaging[24,25]. Indeed, at prostatectomy 44% of patients with high PSA recurrence had high grade GG 4 or 5 cancer, 66% had tumour extension outside the prostate (pT3) and one third had micro-metastasis in the lymph nodes (Supplementary Table 2), while none had received ADT prior to fecal sample collection. Accordingly, these patients have a more biologically aggressive cancer than those in the pre-prostatectomy group who have localized cancer or those with very low PSA who are more likely to only have a local recurrence in the surgical bed. The gut microbiota profiles of these three groups of patients were compared using high-throughput 16S rRNA sequencing (Fig. 1f). First, we observed significant differences at the phylum level in the gut microbiota of patients with BCR compared to no BCR. Noticeably, Firmicutes relative abundance was 21.7 ± 1.5% and 18.1 ± 1.2% higher in patients with high PSA compared to patients with low or undetectable PSA, respectively (Fig. 1f, red bars, $p = 0.04$, two-sided Welch's $t$-test). We also noted that Bacteroidetes relative abundance in BCR high PSA was 91 ± 11% lower compared to no BCR patients (Fig. 1f, green bars, $p = 1.5e^{-6}$, two-sided Welch's $t$-test). The Shannon alpha diversity index was also reduced by 17.9 ± 3.1% and 13.3 ± 4.1% in fecal samples from patients with high PSA compared to patients with undetectable or very low PSA levels, respectively (Fig. 1g, $p = 0.02$, two-sided Welch's $t$-test). Beta-diversity analysis also showed a significant shift in the high PSA population compared to low PSA or no BCR patients (Fig. 1h, $p = 0.03$, Permanova), supporting that patients with

high PSA or early metastatic PCa after prostatectomy display the strongest alterations in their gut microbiota.

**PCa mouse models.** To explore the functional connections between PCa and gut microbes, we recapitulated our observations from PCa patients in two different sets of experiments using syngeneic PCa mouse models to test the effect of tumour volume and cancer aggressiveness on fecal microbiota. In the first, we investigated the effect of tumour volume on the microbiota of the same mice at different time points corresponding to increasing tumour volumes of the SV40 large T antigen-transformed TRAMP-C2 PCa cells injected subcutaneously (s.c.) in the flanks of immunocompetent single-housed

C57BL/6 N syngeneic mice. Fecal samples were harvested at minimal mass formation (palpable tumour), active growth phase and end point, i.e. 2 cm$^3$ total tumour volume (Fig. 2A). When comparing fecal samples from actively growing and end point tumours to no tumour time point samples, 16S rRNA metataxonomic analysis revealed $74 \pm 7\%$ and $71 \pm 8\%$ increase in the Bacteroidetes phylum, respectively (Fig. 2b, $p = 8e^{-4}$, two-sided Welch's $t$-test). As in the human prostate cancer fecal samples, in the TRAMP-2 model we also noted a $14 \pm 5\%$ and $16 \pm 7\%$ reduction in alpha-diversity, measured by the Shanon Index, for the microbiota profiles collected during active growth and at end point compared to the baseline microbiota, respectively (Fig. 2c, $p = 3e^{-4}$, $p = 2e^{-4}$, respectively, two-sided Welch's $t$-test).

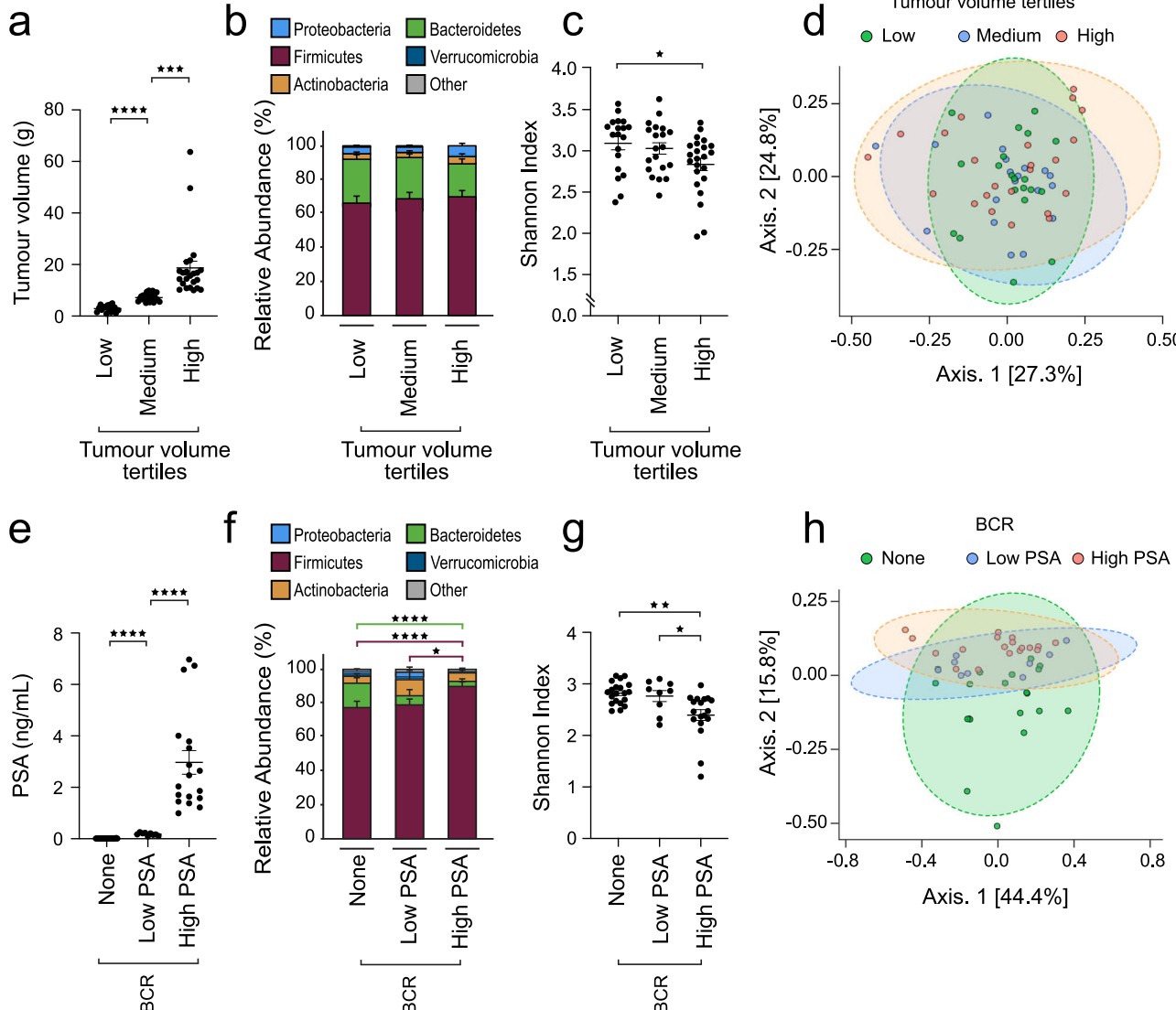

**Fig. 1 | Association of the gut microbiota with tumour volume and aggressiveness in prostate cancer patients. a** Tumour volume from prostate cancer patients treated by radical prostatectomy was estimated based on the % of cancer tissue identified at the pathological examination and the total prostate mass (g) at surgery ($n = 62$ total, low $n = 19$, medium $n = 20$, high $n = 23$). Two-sided Welch's $t$-test was used for comparing groups. **b** Metataxonomic analysis at the phylum level of fecal samples from patients with different prostate tumour burden described in (**a**), fecal samples were harvested $5.0 \pm 0.5$ weeks before surgery. **c** Shannon diversity index at family taxonomic level of 16S rRNA sequences associated with fecal samples from prostate cancer patients with different tumour burden described in (**a**). Two-sided Welch's $t$-test was used for comparing groups. **d** Bray-Curtis beta-diversity analysis at the family taxonomic level and Principal Component

Analysis (PCA) representation for the fecal microbiota corresponding to samples in (**a**). **e** Blood PSA levels from prostate cancer patients recently treated by radical prostatectomy, either without any biochemical recurrence (BCR-None, $n = 20$), with early (Low PSA 0.05–0.49 ng/mL, mean PSA 0.17 ng/mL, SE $\pm$ 0.02, $n = 9$) or definitive BCR (PSA > 0.5 ng/mL, mean 2.96 ng/mL, SE $\pm$ 0.46, $n = 18$). Two-sided Welch's $t$-test was used for comparing groups. **f** Metataxonomic analysis at the phylum level of fecal samples from patients in (**e**). Two-sided Welch's $t$-test was used for comparing groups. **g** Shannon diversity index at family taxonomic level of 16S rRNA sequences associated with fecal samples from prostate cancer patients described in (**e**). Two-sided Welch's $t$-test was used for comparing groups. **h** Bray-Curtis beta-diversity analysis at the family taxonomic level and PCA representation for the fecal microbiota corresponding to sample in (**e**). Graphs are mean $\pm$ SEM.

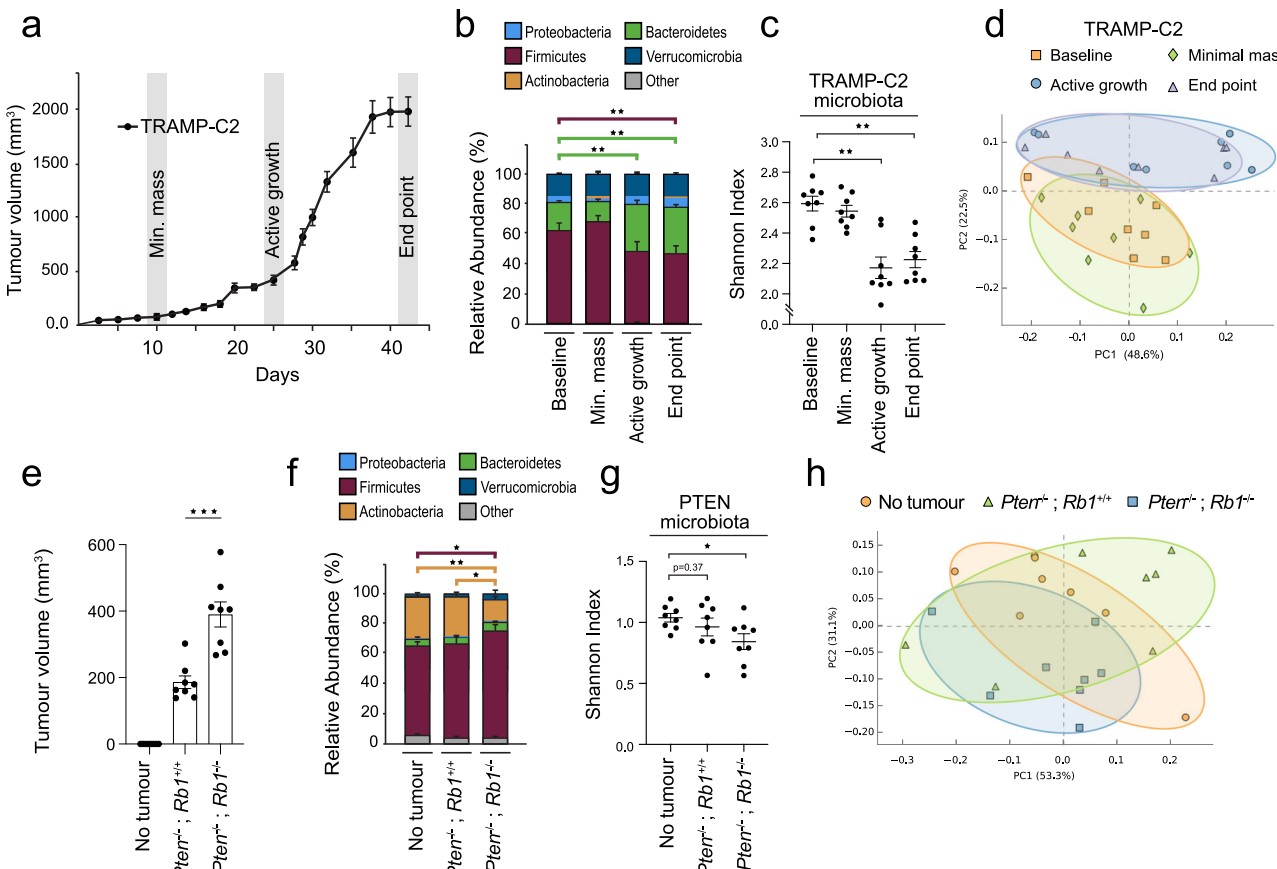

**Fig. 2 | Ectopic prostate tumour development triggers changes in the fecal microbiota in independent syngeneic mouse models. a** Growth phases analysis of TRAMP-C2 tumour cells injected into the flank of immunocompetent single-housed C57BL/6 N mice ($n = 12$/group). **b** Relative abundance of bacterial phyla of fecal samples harvested from tumour-free animals (baseline) and three time points corresponding to minimal masses (first detectable tumour), actively growing and late stage (end point) TRAMP-C2 tumours ($n = 8$/group). $p = 8e^{-4}$, two-sided Welch's *t*-test was used for comparing groups. **c** Shannon diversity index at family taxonomic level of 16S rRNA sequences associated with fecal samples harvested from single-housed C57BL/6 N mice injected with TRAMP-C2 prostate tumour at different development stages ($n = 8$/group). $p = 3e^{-4}$, $p = 2e^{-4}$, for active growth and end point, respectively, two-sided Welch's *t*-test was used for comparing groups. **d** Bray-Curtis beta diversity analysis and PCA visualization of 16S rRNA sequences associated with fecal samples from mice injected with TRAMP-C2 prostate cancer cells and at different phases of tumour development ($n = 8$/group). **e** Tumour volume

analysis at the 4 week time-point of immunocompetent C57BL/6 J mice injected with syngeneic $Pten^{-/-}$ or Pten$^{-/-}$; $Rb1^{-/-}$ prostate tumour cells or control animals without tumour cells ($n = 8$/group). $p = 6e^{-4}$, two-sided Welch's *t*-test was used for comparing groups. **f** Relative abundance of bacterial phyla of fecal samples harvested from tumour-free animals (no tumour) and mice bearing $Pten^{-/-}$ or $Pten^{-/-}$; $Rb1^{-/-}$ prostate tumours ($n = 8$/group) 4 weeks after tumour cell injections. $p = 0.01$ for firmicutes and $p = 0.001$ for actinobacteria, two-sided Welch's *t*-test was used for comparing groups. **g** Shannon diversity index at family taxonomic level of 16S rRNA sequences associated with fecal samples from single-housed tumour-free C57BL/6 J animals (no tumour) and animals with either Pten$^{-/-}$ or $Pten^{-/-}$; $Rb1^{-/-}$ prostate tumours ($n = 8$/group). $p = 0.02$, two-sided Welch's *t*-test was used for comparing groups. **h** Bray-Curtis beta diversity analysis and PCA visualization of 16S DNA sequences associated with fecal samples from mice injected with $Pten^{-/-}$ and $Pten^{-/-}$; $Rb1^{-/-}$ prostate tumour cells and different tumour burden ($n = 8$/group). Graphs are mean ± SEM.

Principal component analysis using the Bray-Curtis beta-diversity index revealed similarities between the fecal microbiota of animals at the stage of active growth and end point (Fig. 2d). In contrast, the beta-diversity of both active growth and end point microbiota were also significantly distinct from those of animals at baseline or with minimal tumour mass (Fig. 2d, $p = 0.001$, Permanova).

In the second set of experiments, we tested the effect of increasing PCa aggressiveness on the gut microbiota collected at about 4 weeks post-injection in three groups of 8 mice each injected *s.c.* with murine PCa cell lines driven either by the loss of *Pten* ($Pten^{-/-}$; $Rb1^{+/+}$) or the loss of both *Pten* and *Rb1* ($Pten^{-/-}$; $Rb1^{-/-}$)[26,27] and control tumour-free mice (Fig. 2e). These experiments were performed in immunocompetent male C57BL/6 J syngeneic mice housed in an independent animal facility under similar diet conditions. The $Pten^{-/-}$; $Rb1^{-/-}$ prostate cancer cells closely resemble the genetic makeup of metastatic prostate cancer cells and were therefore used to emulate our previous findings in patients with high PSA BCR[28,29]. As expected, $Pten^{-/-}$; $Rb1^{-/-}$ tumours grew significantly larger than $Pten^{-/-}$; $Rb1^{+/+}$

tumours at the 13-week time point (Fig. 3e). Compared to controls, at the phylum level, we observed a 20 ± 5% increase of Firmicutes relative abundance similar to PCa patients with high PSA BCR ($p = 0.01$, two-sided Welch's *t*-test) and a 48 ± 14% reduction of Actinobacteria relative abundance ($p = 0.001$, two-sided Welch's *t*-test) in fecal samples of animals with $Pten^{-/-}$; $Rb1^{-/-}$ tumours but not in the animals with $Pten^{-/-}$; $Rb1^{+/+}$ tumours (Fig. 2f). We observed that the Shannon index of the microbiota of $Pten^{-/-}$; $Rb1^{-/-}$ mice was reduced by 19 ± 8% compared to mice without tumours ($p = 0.02$, two-sided Welch's *t*-test), but not for $Pten^{-/-}$; $Rb1^{+/+}$ tumour bearing mice (Fig. 2g). In addition, the microbiota from animals with $Pten^{-/-}$; $Rb1^{-/-}$ tumours were significantly different from the microbiota of no tumour controls using the Bray-Curtis beta-diversity principal component analysis (Fig. 2h, $p = 0.02$, Permanova), similar to the same analysis in the TRAMP-C2 model (Fig. 2d). Hence, the strongest loss of gut microbiota alpha diversity was observed in the largest tumours within the same TRAMP-C2 model and the more aggressive $Pten^{-/-}$; $Rb1^{-/-}$ tumours. These results support our observations in the two PCa patient cohorts that tumour volume,

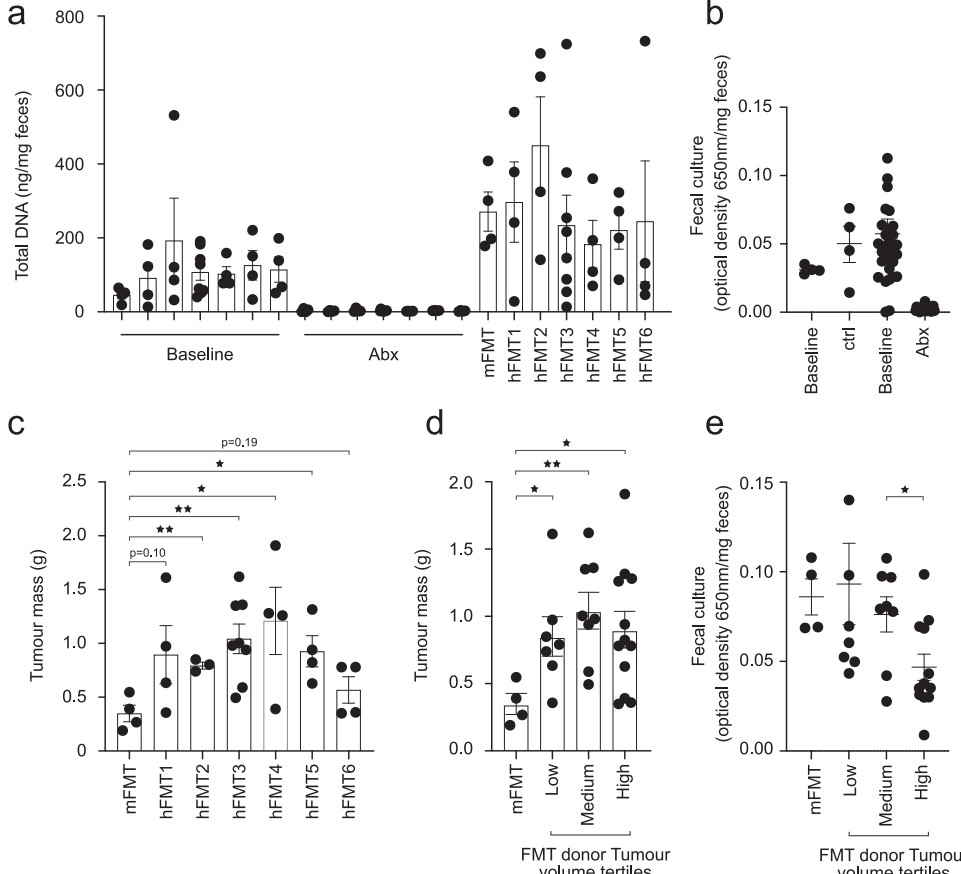

**Fig. 3 | Causal impact of the gut microbiota on prostate cancer. a** Total DNA was extracted from feces of C57BL/6 N mice at baseline, after 1 week of antibiotic (Abx) treatment and after series of two fecal microbiota transplants (FMT). Microbiota depletion was performed by Abx treatment and mice were rescued with their pre-Abx fecal microbiota transplant (mFMT, $n = 4$ mice) or transplanted with fecal samples from six independent patients with grade group ≥2 localized prostate cancer ($n = 4$ mice/group for hFMT1, 2, 4, 5, 6 and $n = 8$ mice for hFMT3). The hFMT3 was repeated twice in two different experiments. **b** Gifu Anaerobic Media (GAM) broth cultures of fecal samples from mice before (baseline) and after 4 days of Abx treatments (Abx, $n = 16$), controls were mice untreated ($n = 4$). **c** C57BL/6 N Abx-pretreated mice received two FMT from 6 independent patients with localized prostate tumor. After the FMTs, mice were injected with TRAMP-C2 prostate tumour cells and tumour mass was measured at sacrifice, 5 weeks after tumour cells injections ($n = 4$ mice/group for hFMT1, 4, 5, 6, $n = 3$ mice for hFMT2 and $n = 8$ mice for hFMT3). $p = 0.006$ (hFMT2), $p = 0.007$ (hFMT3), $p = 0.04$ (hFMT4) and $p = 0.01$ (hFMT5), Student's t-test. **d** The data in (**c**) was compiled based on the donor's tumor volume at the pathological analysis and corresponding to the tertiles defined in Fig. 1a ($n = 4$ mice for mFMT, $n = 7$ low tumour volume hFMTs, $n = 8$ medium tumour volume hFMTs and $n = 12$ High tumour volume hFMTs), $p = 0.04$ (Low), $p = 0.007$ (Medium) and $p = 0.04$ (High), Student's t-test. **e** GAM broth cultures of fecal samples from mice before corresponding to (**d**) ($n = 4$ mice for mFMT, $n = 7$ low tumour volume hFMTs, $n = 8$ medium tumour volume hFMTs and $n = 12$ High tumour volume hFMTs), $p = 0.03$, two-sided Welch's t-test. Graphs are mean ± SEM.

and especially cancer aggressiveness, are linked to the strongest fecal microbiota alterations (Fig. 1c, g).

### PCa patients gut microbiota drives TRAMP-C2 tumour growth

To test whether the gut microbiota influence PCa growth, we used fecal microbiota transplantation (FMT) of human feces collected pre-prostatectomy from 6 patients with Grade Group 2 to 5 PCa and distinct tumour volumes at surgery (Supplementary Table 3) in the syngeneic TRAMP-C2 mouse model. Animals were first depleted of their microbiota using antibiotics given orally. Antibiotics effectively depleted gut microbiota as measured by total DNA extracted from fecal samples (Fig. 3a) and Gifu anaerobic broth cultures from fecal samples (Fig. 3b). After a 48 h wash-out, mice were inoculated with the six different human fecal suspensions and a control mouse homologous tumour-free FMT that was harvested before the antibiotic regimen ($n = 4$/FMT except for FMT3 repeated once $n = 8$). After two series of FMT, TRAMP-C2 PCa cells were injected s.c. in both flanks of the mice and tumour mass (g) was assessed at 5 weeks. While human PCa FMT1 and FMT6 showed a trend to increase (734 ± 30% and 63 ± 22%, respectively) TRAMP-C2 tumour mass compared to

autologous mouse FMT, FMT2, 3, 4 and 5 all led to bigger tumours (127 ± 4% $p = 0.006$, two-sided t-test, 199 ± 13% $p = 0.007$, two-sided t-test, 247 ± 26% $p = 0.04$, two-sided t-test and 166 ± 15% $p = 0.01$, two-sided t-test, respectively) in the recipient animals (Fig. 3c). However, there was no trend for increased TRAMP-C2 volume with FMTs from patients with increasing tumour volume (Fig. 3d). On the other hand, we observed 39 ± 16% less viable bacteria in fecal cultures of mice having received FMTs from patients with high tumour volume and low microbiota diversity (Fig. 1c) compared to medium tumour volume FMTs (Fig. 3e, $p = 0.03$, two-sided Welch's t-test). This suggests that the low diversity of the microbiota could potentially impair fecal transplantation and thus explain the lack of increased TRAMP-C2 tumour volume for FMTs derived from high tumour volume patients. Overall, these results support a crosstalk by which the gut microbiota composition also influences PCa growth.

### PCa alters microbiota-host metabolism by modulating specific microbes

We further explored if specific microbes were systematically altered in fecal samples of PCa-bearing mice. We observed that the levels of

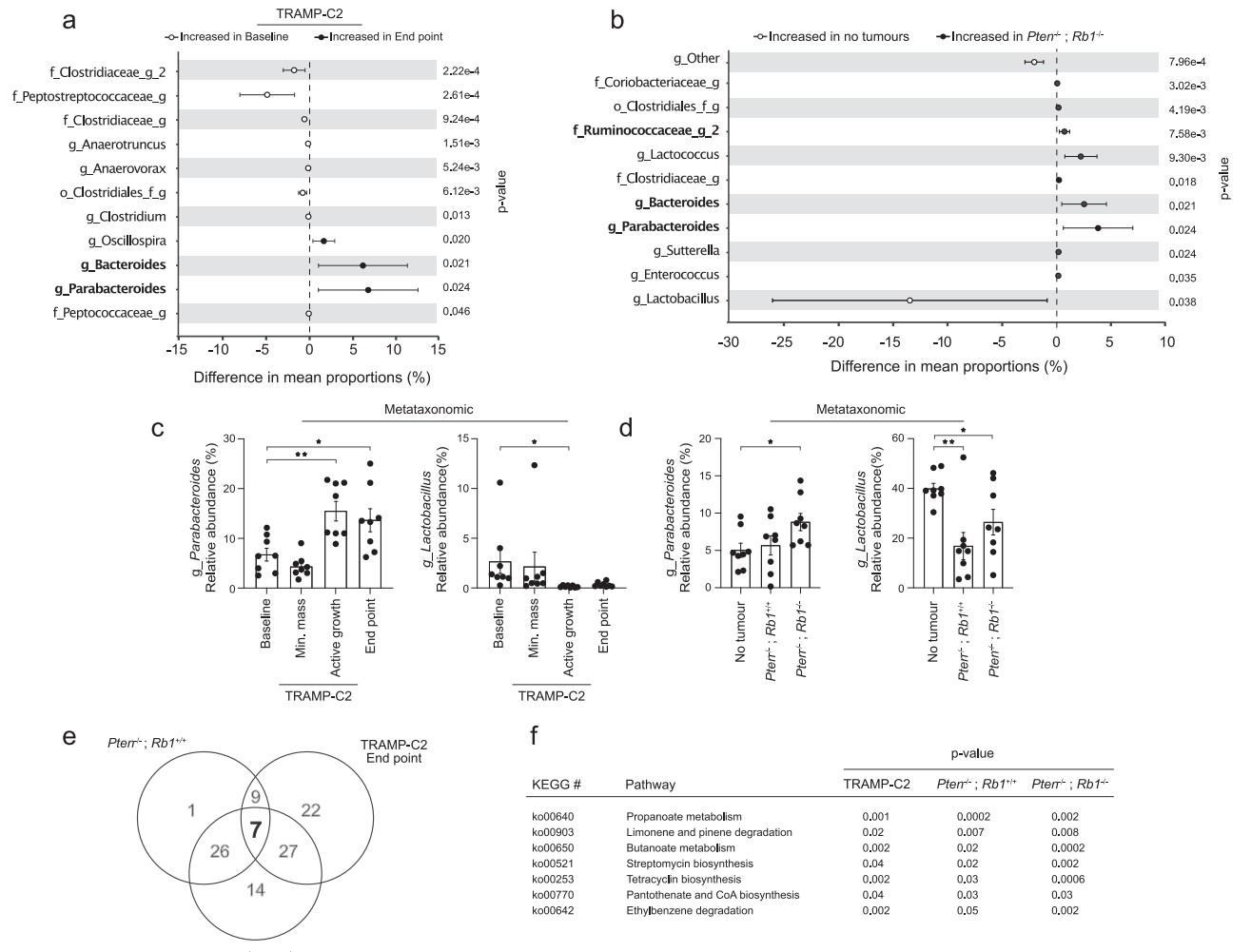

**Fig. 4 | Common fecal 16S rRNA-related signatures in three independent models of prostate cancer. a** Differential enrichment of bacterial genera from mice with TRAMP-C2 tumours at the end point growth stage (late tumour growth) and the baseline before tumour cell injection (*n* = 8 mice/group). **b** Differential enrichment of bacterial genera corresponding to mice with *Pten*−/−; *Rb1*−/− tumours and tumour-free mice (*n* = 8 mice/group). **c** Relative abundance, in percentage (%) of total 16S rRNA sequences, of *Parabacteriodetes* 16S rRNA (left, *p* = 0.003 (Active growth) and *p* = 0.02 (End point), Welch's *t*-test) and *Lactobacillus* 16S rRNA (right, *p* = 0.05 (Active growth), Welch's *t*-test) from fecal samples of mice with TRAMP-C2 tumour harvested at different stages of prostate cancer development (*n* = 8 mice/ group). **d** Relative abundance, in % of total 16S rRNA sequences, of *Para-bacteriodetes* 16S rRNA (left, *p* = 0.02, Welch's *t*-test) and *Lactobacillus* 16S rRNA

(right, *p* = 0.003 (*Pten*−/−; *Rb1*+/+) and *p* = 0.04 (*Pten*−/−; *Rb1*−/−), Welch's *t*-test) from fecal samples of mice without tumour, with *Pten*−/−; *Rb1*+/+ or *Pten*−/−; *Rb1*−/−, tumours (*n* = 8 mice/group). **e** Ven diagram of a PICRUSt analysis of functions predicted to be enriched in fecal samples from TRAMP-C2 tumours at end point compared to baseline controls. The enrichment of predicted function was also performed for *Pten*−/−; *Rb1*+/+ and *Pten*−/−; *Rb1*−/− tumours and compared to no-tumour controls is also shown. **f** The 7 predicted functions between fecal tumour samples and controls that were commonly enriched between mice bearing TRAMP-C2, *Pten*−/−; *Rb1*+/+ and *Pten*−/−; *Rb1*−/− tumours, following a PICRUSt analysis are displayed. Graphs in (**a**, **b**) are mean ± SD. Graphs in (**c**, **d**) are mean ± SEM, Welch's *t*-test.

several genera were differentially affected in phases of end point tumour growth for TRAMP-C2 (Fig. 4a, Fig. S1A) and both *Pten*−/−; *Rb1*+/+ and *Pten*−/−; *Rb1*−/− (Fig. 4b, Fig. S1B) mouse models. For the late-stage TRAMP-C2 and *Pten*−/−; *Rb1*−/− tumour bearing animals, we noted 6.3 ± 2.7% and 6.9 ± 3.0% increase in mean proportion difference of Bacteroides, together with 2.5 ± 1.0% and 3.8 ± 1.6% increase in the mean proportion difference of Parabacteroides fecal DNA sequences, respectively (Fig. 4a, b). Interestingly, we observed 3.5 ± 1.7% increase of fecal Ruminococcaceae_g_2 sequences in mice with TRAMP-C2 minimal masses compared to baseline samples (Fig. S1A). We also noted 0.7 ± 0.2% and 0.5 ± 0.2% Ruminococcaceae_g_2 increase in both *Pten*−/−; *Rb*−/− (Fig. 4b) and *Pten*−/−; *Rb1*+/+ fecal samples compared to no tumour controls (Fig. S1B). We next proceeded to validate a portion of the 16S rRNA sequencing results and selected two genera that were found either enriched or depleted from both TRAMP-C2 and *Pten*−/−; *Rb1*−/− fecal samples (Fig. 4c, d). The metataxonomic results matched

the qPCR results for the two specific selected species corresponding to Parabacteroides and Lactobacillus genera; Parabacteroides distasonis and Lactobacillus murinus, respectively (Fig. S2C). Changes in SV40 large T antigen-driven tumour-associated microbiota were more resembling with *Pten*−/−; *Rb1*−/−-related microbiota than *Pten*−/−; *Rb1*+/+, again supporting that *Rb1* loss of function is prompting the growth of larger tumours resulting in more significant gut microbiota alterations. Taken together, our results highlight common gut microbiome alterations in relation to tumour development in three independent mouse models of PCa, suggesting that specific gut microbes might drive the loss of alpha diversity and participate in the tumour development process.

To identify a molecular underpinning between the tumour-associated gut microbiota alterations and the host, we used Phylogenetic Investigation of Communities by Reconstruction of Unobserved States (PICRUSt)[30] to extrapolate functions associated with 16S rRNA

sequencing data. PICRUSt analyses outlined 65, 43 and 74 functions altered in fecal samples from mice with end point TRAMP-C2 tumours, Pten[-/-]; Rb1[+/+] and Pten[-/-]; Rb1[-/-] tumours compared to their corresponding no tumour controls, respectively (Fig. 4e, Fig. S2A–E). We found only three different functions altered in fecal samples of TRAMP-C2 minimal mass animals compared to baseline (Fig. S2D). To the contrary and consistent with our observations with the 16S rRNA profiling data, we observed significantly more predicted functions shared between TRAMP-C2 tumours-associated microbiota and Pten[-/-]; Rb1[-/-]-related microbiota compared to Pten[-/-]; Rb1[+/+]-related microbiota. Interestingly, compared to the gut microbiota of tumour-free mice, we found that seven predicted Kyoto Encyclopedia of Genes and Genomes (KEGG) metabolic pathways were commonly altered in the gut microbiota of mice for all the end point PCa tumours TRAMP-C2, Pten[-/-]; Rb1[+/+] and Pten[-/-]; Rb1[-/-] mice models (Fig. 4f). We next manually screened the components of these seven KEGG pathways and pinpointed three enzymes common to three of these seven altered pathways: Enoyl-CoA hydratase (Paaf/echA), Aldehyde dehydrogenase (ALDH) and 3-hydroxyacyl-CoA (FadB). These enzymes are normally involved in the metabolism of fatty acids, namely aerobic and anaerobic degradation of long-chain Fatty Acids (LCFA) via beta-oxidation cycle. Overall, these results suggest that PCa alters microbiota-host metabolism by modulating distinct and common host fecal bacteria involved in the metabolism of LCFA.

## Omega-3 FA supplementation modulates the gut microbiome and PCa growth and aggressiveness

PCa incidence and progression are noticeably affected by modifiable lifestyle factors including diet. For example, several studies have linked animal-derived saturated fat uptake with PCa incidence[31,32] and development[33,34]. Diet rich in omega-3 fatty acids, such as the Mediterranean diet, was shown to reduce PCa death whereas diet rich in omega-6 such as the typical North-American diet increases the risk. Long-chain polyunsaturated fatty acids (PUFA), such as omega-3 fatty acids, were previously reported to cause whole-body metabolic changes that were dependent on gut microbiota[35,36]. We hypothesized that a targeted PUFA dietary intervention could potentially alter the gut microbiota induced metabolism of fatty acid affecting PCa growth. To test this hypothesis, C57BL/6 N mice were supplemented with different purified monoglyceride (MAG) PUFAs: an omega-6 (MAG-AA), two omega-3 (MAG-DHA, MAG-EPA) and an omega-9 (HOSO) molecule commonly used as control in human clinical trials. After two weeks of gavage, animals were injected s.c. with TRAMP-C2 cells to test the impact of each single PUFAs on PCa growth. We observed a decrease of $45 \pm 10\%$ of the TRAMP-C2 tumour growth in MAG-DHA-fed animals compared to control HOSO-treated TRAMP-C2 tumours ($p = 0.04$, Student's t test) (Fig. 5a), and a similar trend for MAG-EPA-fed mice. Next, we profiled the fecal microbiome of these animals at sacrifice using high throughput 16S rRNA metataxonomic analysis. Interestingly, the abundance of sequences related to the Ruminococcaceae family (Fig. 5b, c) in the two omega-3 PUFAs MAG-DHA- and MAG-EPA-treated animals were reduced by $52 \pm 6\%$ and $43 \pm 9\%$ respectively ($p = 0.008$, two-sided Welch's t-test) compared to HOSO control. Of note, the baseline fecal microbiome profiles and Ruminococcaceae levels were not different between the four groups (Fig. S3A, B). These results, under strictly controlled experimental conditions, support an interplay between the dietary omega-3 PUFAs, the gut microbiota and PCa growth.

We further tested the potential effect of long chain omega-3 polyunsaturated fatty acids (LCn3) supplement on gut Ruminococcaceae and PCa aggressiveness in 41 PCa patients enrolled in a pre-radical prostatectomy trial[21,37] who provided fecal samples before and after the intervention. These samples are a sub-cohort of the last 41 patients from a parental randomized clinical trial of 130 PCa patients where the effect of a purified MAG-EPA supplement was compared to placebo

(HOSO) taken orally daily for an average of 7.2 weeks (±0.37) prior to radical prostatectomy (NCT02333435)[21]. The clinical and biopsy characteristics at baseline and the pathological characteristics at radical prostatectomy of the sub-cohort of 41 PCa patients who agreed to provide fecal samples for research are shown in Supplementary Tables 4, 5, respectively.

PCa grade is one of the most important determinants of PCa aggressiveness and risk of progression. However, PCa grading is complex due to the frequent multifocality of tumours with distinct aggressive potential within an individual prostate[38]. The new International Society of Urological Pathology (ISUP) system of five prostate cancer Group-Grades (GG) is based on the proportion of the three Gleason patterns 3, 4 or 5[39–41]. GG 1 tumours have Gleason pattern 3 only, while GG 2, 3 and 4 are defined by the proportion of pattern 4 (<50%, >50% and 100% respectively), and GG 5 have pattern 5. This new classification showed less variation between the grade of the diagnostic biopsy and that of the prostatectomy specimen than the traditional Gleason score[42,43]. Previous studies showed that patients with up-graded disease exhibited more aggressive pathological features than concordant tumours and a higher risk of biochemical recurrence after radical prostatectomy[44].

When comparing the cancer grade on the diagnostic biopsy with the grade on the prostatectomy specimen in the 41 patients, we observed an up-grading in 8 patients, a down-grading in 5 patients and no change in the other 28 patients. We first compared the proportion of Ruminococcaceae in all fecal samples of patients before and after supplement treatment in relation with the PCa GG reclassification. Interestingly, we observed $27 \pm 3\%$ enrichment in Ruminococcaceae relative abundance for patients with PCa upgrade compared to patients with stable GG ($p = 0.004$, two-sided Welch's t-test) (Fig. 5d). Fortunately, the sub-group of 41 patients who provided fecal samples were evenly distributed between the placebo and the MAG-EPA and also showed 71% less GG up-grading with MAG-EPA compared to HOSO ($p = 0.016$, Chi square test) (Fig. 5e). Noticeably, compared to HOSO, MAG-EPA was associated with $12 \pm 5\%$ reduction in the abundance of sequences corresponding to the Ruminococcaceae ($p = 0.046$, paired-Wilcoxon test) (Fig. 5f) between pre- and post-treatment microbiota.

Several microbiota-derived molecules are known to directly affect host's metabolism[45]. For example, SCFA are produced by commensal bacteria, including members of the Ruminococcaceae family and interactions between SCFA and human cancers have been described before[46,47]. We therefore measured the fecal levels of a panel of SCFA in the 41 patients before and after MAG-EPA or HOSO treatment. Compared to placebo, MAG-EPA reduced fecal butyric acid levels by a factor of 1.5 on average ($p = 0.04$, two-sided Welch's t-test). There was no effect on other SCFAs (Fig. 6a–f), except for a similar reduction trend of valeric acid levels (Fig. 6d). These observations were also supported by a qPCR analysis of the different enzymes responsible for the synthesis of SCFA including butyric acid (Fig. S4A, B). Patients with PCa downgrading at surgery displayed less enrichment on average by a factor of 2 of fecal butyric acid levels than patients with no change ($p = 0.002$, two-sided Welch's t-test) (Fig. 6g). Of note, 4 of the 5 patients with downgrading received MAG-EPA pre-surgery. Patients with PCa upgrade showed a trend for higher butyrate fold change over time compared to downgraded patients ($p = 0.07$, two-sided Welch's t-test) (Fig. 6g). To further explore the potential association of increase fecal butyrate with PCa aggressiveness we measured the fecal butyric content in the cohort of PCa patients with or without PSA failure after prostatectomy from Fig. 1e–h. Fecal butyric acid concentrations were $72 \pm 14\%$ higher in patients with early metastasis post-prostatectomy (high PSA BCR, $p = 0.02$, two-sided Welch's t-test) corresponding to Fig. 6h, supporting that PCa aggressiveness also translate into an altered microbiota metabolism. Taken together, our results support that modifiable lifestyle factors such as diet, in particular LCn3, can

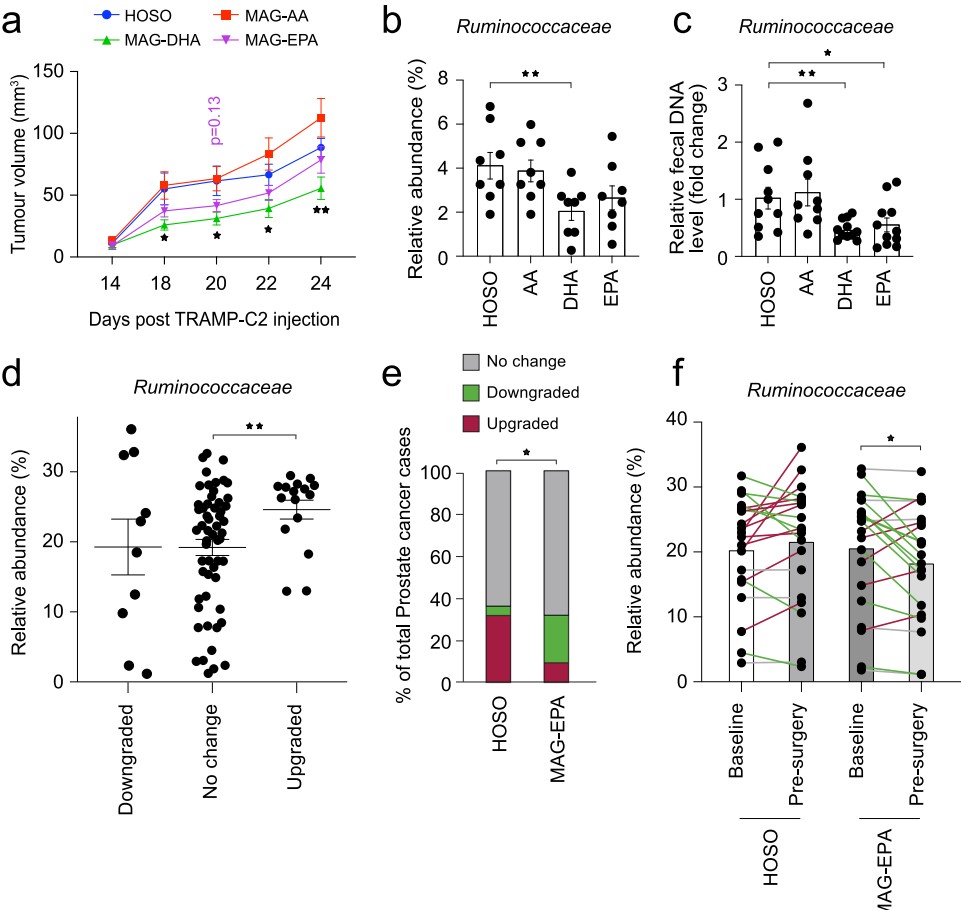

**Fig. 5 | Functional interactions between long-chain polyunsaturated fatty acid, gut microbiota and prostate cancer development. a** TRAMP-C2 tumour growth in mice pre-treated with different purified versions of long-chain polyunsaturated fatty acids (PUFA); monoglyceride arachidonic acid (MAG-AA), MAG-docosahexaenoic acid (DHA) or MAG-eicosapentaenoic acid (EPA) and control high oleic sunflower oil (HOSO) (*n* = 12/group). *p* ≤ 0.05, Welch's *t*-test. **b** Relative abundance, in % of total 16S rRNA sequences, of *Ruminococcaceae* 16srRNA from fecal DNA samples of mice with TRAMP-C2 pre-treated with different PUFA molecules (*n* = 8/group). *p* ≤ 0.05, Welch's *t*-test. **c** qPCR validation of *Ruminococcaceae* 16S rRNA levels from fecal DNA samples of mice with TRAMP-C2 pre-treated with PUFA molecules. Data is presented as relative levels (fold change) normalized to total 16S rRNA sequences (*n* = 12/group). *p* ≤ 0.05, Welch's *t*-test. **d** Relative abundance of *Ruminococcaceae* from fecal DNA samples of patients in relation to their prostate tumour grade change (Gleason score from histopathological analysis of prostate tumour at surgery compared to histological score of the biopsy of study enrolment), *p* ≤ 0.05, Welch's *t* test. **e** A randomized clinical trial (NCT02333435)

was performed at our clinical facility testing the effect of MAG-EPA PUFA on prostate cancer patients before radical prostatectomy. A subset of 41 patients donated fecal samples for research at study baseline and the morning before surgery, 7.2 ± 0.37 weeks later. Change in prostate tumour grade was compared between patients receiving MAG-EPA or placebo 7 weeks prior to surgery. The percentage of patients without any change in their cancer grade, with apparent downgrade or upgrade in their prostate cancer grade group score is shown as a fraction of the total cases per group. Statistical test was Chi square comparing % for the 3 categories between MAG-EPA (*n* = 21) and placebo control (*n* = 20). *p* ≤ 0.05, chi square test. **f** Using DNA extracted from fecal samples corresponding to (**e**), we used 16S rRNA metataxonomic to compare the profiles corresponding to men before and after 7.2 ± 0.37 weeks of MAG-EPA (*n* = 21) supplementation or placebo (*n* = 20). The relative abundance of sequences corresponding to *Ruminococcaceae* is shown, *p* ≤ 0.05, Paired–Wilcoxon test. Line colors represent *Ruminococcaceae* enrichment (red), depletion (green) or no change (grey). Graphs are mean ± SEM.

modulate the crosstalk between gut microbes, their metabolites and PCa (Fig. 7).

## Discussion

Several research groups observed imbalances in the gut microbiota of patients with cancers including PCa compared to healthy individuals[1–3], but most studies were in patients treated by immunotherapy or ADT in the case of PCa. Yet, the potential interplay between cancer development and changes in gut microbiota diversity has received little attention. In the present study we found a systematic reduction in gut microbiota diversity correlating with increasing PCa volume and aggressiveness in two distinct clinical scenarios of patients without ADT. The great majority of patients with an average PSA of 2.96 ng/ml after prostatectomy have metastases in lymph nodes and/or bones that can now be observed by highly sensitive TEP-PSMA imaging.

Patients with the highest cancer volume in the prostatectomy specimen also had a significantly reduced gut microbiota diversity. These observations are reminiscent of the correlation found in pancreatic cancer patients between reduced gut microbiota alpha diversity and shorter-term patient's survival[48]. We also observed reduced gut microbiota diversity in the two distinct mouse models that grew the largest tumours, in particular those with loss of both *Pten* and *Rb1*, an alteration frequently observed in metastatic PCa patients[28,29].

It is not clear how PCa could cause steady-state changes in gut microbiota since the prostate and the intestine are anatomically separated. Moreover, the facts that a more important impact on gut microbiota was observed in patients with early metastasis (High PSA BCR) and no prostate, as well as in our mouse models of subcutaneously implanted PCa, point towards a mechanism involving prostate tumour-secreted molecules that could mediate changes in

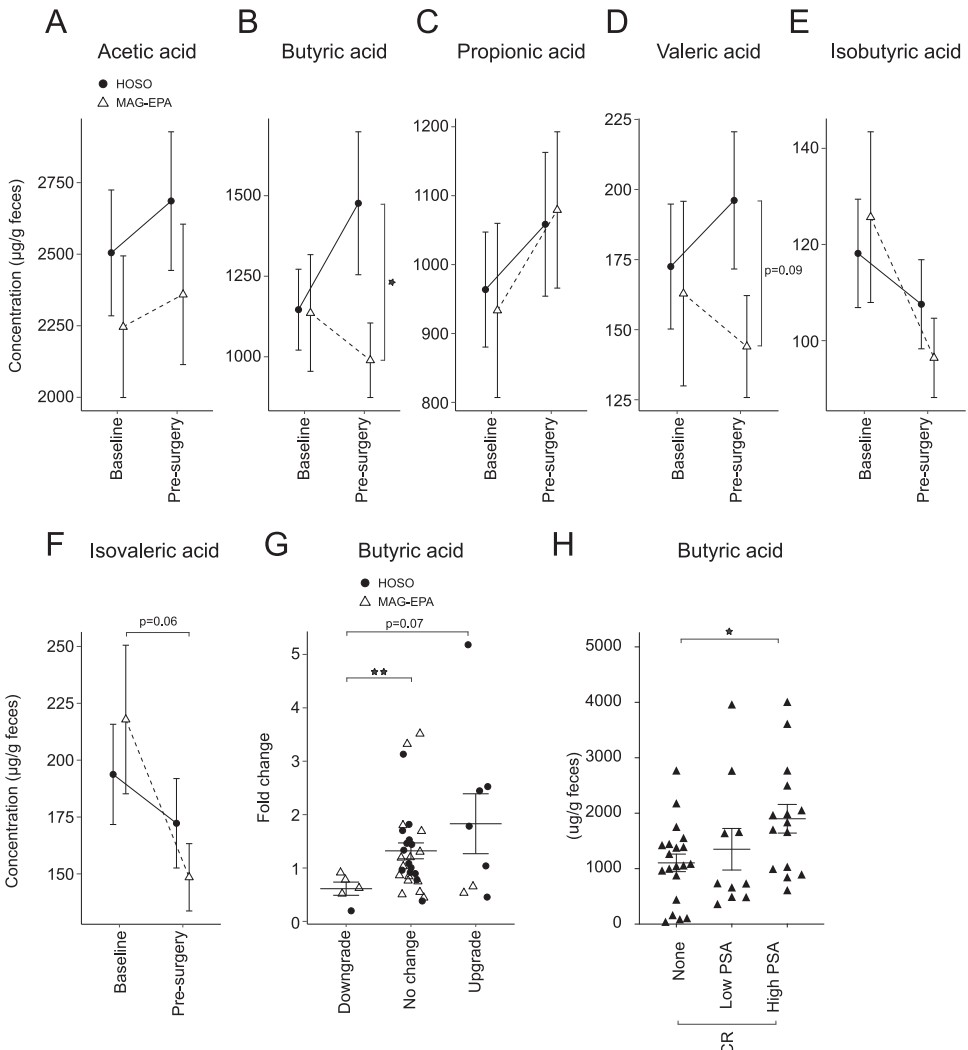

**Fig. 6 | A dietary intervention reduces fecal short-chain fatty acid (SCFA) levels in prostate cancer patients.** Gas chromatography coupled with flame ionization detection (GC-FID) was used to measure the levels of different SCFA from feces of prostate cancer patients at baseline and after daily supplementation of mono-glyceride eicosapentaenoic acid (MAG-EPA) PUFA ($n = 21$) or control of high oleic sunflower oil (HOSO) ($n = 21$) for $7.2 \pm 0.37$ weeks before radical prostatectomy. The different SCFAs measured were; (**a**) Acetic acid, (**b**) Butyric acid ($p = 0.04$, Welch's $t$-test). **c** Propionic acid, (**d**) Valeric acid, (**e**) Isobutyric acid, (**f**) Isovaleric acid.

**g** Fecal butyric acid levels were compared between baseline and pre-surgery for prostate cancer patients ($n = 41$). Fold change in butyric acid levels was associated with downgrade, no change or an upgrade of prostate cancer between radical prostatectomy and enrolment biopsy ($p = 0.002$, Welch's $t$-test). **h** Fecal butyric acid levels were measured in patient samples corresponding to Fig. 1e and presented in relation to PSA biochemical recurrence (BCR) status ($p = 0.02$, Welch's $t$-test). Graphs are mean ± SEM.

the gut microbiota homeostasis. For example, members of the kallikrein (KLK) family of peptidases produced by the prostate are known to participate in maturation and activation of antimicrobial peptides[49–52]. PSA (KLK3), is secreted by normal and PCa cells and its serum levels are increased with increasing PCa burden. While PSA is not expressed by murine prostate cells, different homologous proteins secreted by the murine prostate such as α-microseminoprotein (PSP94) have reported anti-bacterial activity[53]. Moreover, a correlation was observed between the number of culturable bacteria isolated from prostate samples and serum PSA levels in patients with prostate hyperplasia who underwent transurethral prostate resection[54].

We used three mouse models, housed in separate animal facilities to identify common gut microbe alterations as a consequence of PCa development. Under these controlled conditions, we observed significant alterations in the composition of microbiota as a function of tumour growth stage. Interestingly, we found the most prominent changes in gut microbiota diversity and composition from fecal samples of animals with end point TRAMP-C2 tumours and tumours with

combined *Pten* and *Rb1* deletions. SV40 large T antigen transgenic expression in TRAMP-C2 prostate tumour cells alters the *Trp53* and *Rb1* signaling pathways leading to constitutive tumour cell growth[55]. Similarly, loss of *Pten* alone or in combination with *Rb1* knockout also triggers PCa cell's constitutive activation of cell division[26]. Interestingly, tumours with combined *Pten* and *Rb1* deletion and TRAMP-C2 tumours caused greater changes in the gut microbiota of recipient animals compared to tumours characterized by the loss of *Pten* alone. Loss of both *PTEN* and *RB1* are also common in patients' metastatic PCa[28,29]. Several hypotheses could explain how the loss of *Rb1* increases the crosstalk between prostate tumour cells and gut microbiota. The state of PCa cell division and metabolic activity could be important for the systemic effect of developing tumour cells towards the gut microbiota. The lesser loss of microbiota alpha diversity in *Pten*[−/−]; *Rb1*[+/+] compared to the TRAMP-C2 and *Pten*[−/−]; *Rb1*[−/−] mouse models suggests that genetic alterations within incipient tumours cells could contribute to cancer-gut microbiota interaction. A significant loss of alpha diversity was also associated with greater tumour volume for

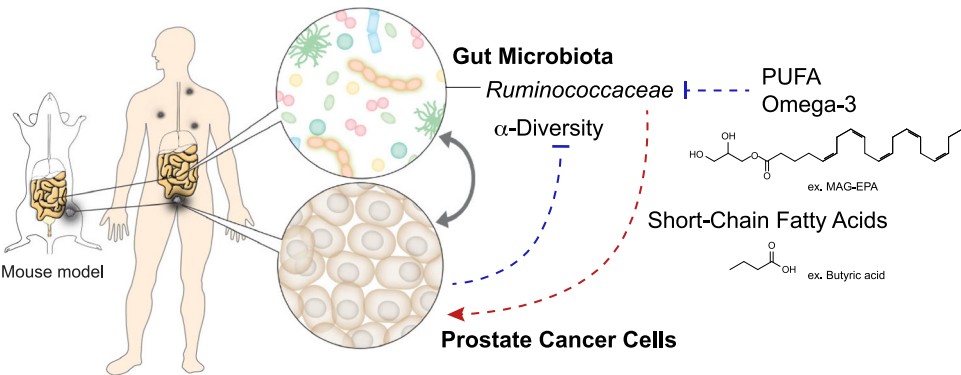

**Fig. 7 | Proposed mechanism of gut microbiome-prostate cancer crosstalk.** Schematic representation for the proposed interaction between dietary long-chain polyunsaturated fatty acids (PUFA) omega-3, the gut microbiota and prostate cancer.

human pre-radical prostatectomy samples. Although we observed a consistent loss of alpha diversity in patients and mouse models of PCa, we did not find a conserved set of taxa systematically altered in PCa samples, except for a significant increase in Firmicutes in fecal samples from early metastatic PCa (High PSA BCR) and in the *Pten*−/−; *Rb1*−/− mouse tumours. Patient-to-patient variability is a common feature of human gut microbiota studies and is not unique to our study[56,57]. As previously reported[58], the baseline microbiota was also different between mouse models, which also possibly impaired our ability to identify consistent changes in gut taxa as a result of PCa development. Repeated measures, paired analysis and whole-genome sequencing could help improve our capacity to identify consistent changes in gut taxa as a result of PCa development. Overall, our results support that PCa growth and aggressiveness are linked to changes in the composition of gut microbiota and our work suggests that genetic alterations within incipient PCa cells might contribute to this gut microbiota-cancer interaction. Although the canonical function of *Rb1* gene product is mainly associated with cell cycle regulation, it was recently found that *Rb1* also regulates the expression of immune checkpoint inhibitors[59,60] suggesting yet another potential immune-mediated mechanism.

Transplantation to mice of human fecal microbiota (FMT) from patients with high tumour volume promoted the growth of the TRAMP-C2 tumours suggesting a bi-directional PCa and gut microbiota crosstalk. Several microbiota-derived molecules are known to directly affect host's metabolism[45]. To investigate further possible metabolic interactions between the tumour-driven gut microbiota alterations and the hosts, we used PICRUSt analyses to extrapolate functions associated with our 16S rRNA sequencing data of gut microbiota from two distinct mouse models at different time points. Compared to that of their tumour-free mice, we found seven predicted KEGG metabolic pathways commonly altered in all large tumours. Further manual screening of components of these KEGG pathways pinpointed three enzymes normally involved in the metabolism of fatty acids, namely aerobic and anaerobic degradation of LCFA. These results suggest that PCa alter microbiota-host metabolism by modulating distinct and common host fecal bacteria involved in LCFA metabolism.

Long-chain polyunsaturated fatty acids (PUFA), such as LCn3, were previously reported to cause whole-body metabolic changes that were dependent on the gut microbiota[35,36]. Previous work suggested that PUFA were capable of modulating PCa in human and similarly in mouse models[61]. PUFA also modulate whole-body metabolism by altering the composition of gut microbiota or via the production of metabolites derived from commensalism[62–64]. Importantly, FMT experiments showed that the transfer of PUFA-derived microbiota to recipient mice was sufficient to recapitulate the effects of PUFA on the host's metabolism[35]. We hypothesized that a targeted PUFA dietary

intervention could potentially alter the gut microbiota induced metabolism of fatty acid affecting PCa growth.

Indeed, our studies showed that TRAMP-C2 tumour growth was reduced in mice fed with LCn3 supplements MAG-DHA and MAG-EPA compared to LCn6 and controls. It was also associated with decreased fecal levels of Ruminococcaceae. In PCa patients treated by prostatectomy, high levels of fecal Ruminococcaceae were associated with cancer upgrading in the prostatectomy specimen compared to the diagnostic biopsy. Treatment of patients for 7 weeks on average before prostatectomy with a LCn3 MAG-EPA supplement compared to placebo was associated with a reduction of cancer upgrading and of fecal Ruminococcaceae levels. Clinical studies conducted on healthy individuals have reported different beneficial impacts of short-term omega-3 supplementation on the gut microbiota based on improved alpha diversity indexes or the transient augmentation of beneficial bacteria such as Bifidobacterium or Lactobacillus[62,65,66]. However, here we report the effect of omega-3 supplements on the gut microbiota of PCa patients and mouse models associated with potentially beneficial effect on cancer. Some studies have reported bactericidal activity for EPA omega-3 supplements towards opportunistic pathogens and bacteria responsible for periodontal disease[67,68]. Ruminococcaceae ecological niche is within the mucosal biofilm layer in the gut[69]. Therefore, by blocking biofilm-producing bacteria, MAG-EPA treatment could have indirectly dislodged endogenous Ruminococcaceae populations. Several reports underlined members of the Ruminococcaceae family that were associated with improved responses to cancer immunotherapy with anti-PD-1 antibodies[9,70]. The gut microbiota was shown to affect the dynamic of circulating immune cells independently of immunotherapy treatments[71,72], supporting that commensal bacteria can educate the immune system and modulate its anti-cancer activity. Likewise, it is increasingly recognized that some bacterial groups produce metabolites that have systemic immune properties, notably the SCFA that play a pivotal role on immune modulation[46]. Here, we observed reduced fecal butyrate levels in the MAG-EPA-treated PCa patients in association with significantly less cancer upgrading at prostatectomy. Conversely, we also observed a significant increase in fecal butyrate levels in early metastatic PCa patients post-prostatectomy (High PSA BCR) compared to those without recurrence suggesting an association with increasing PCa aggressiveness. Butyrate can exert both tumour promotive and suppressive effects and these were more extensively studied in colorectal cancer. While gut epithelial cells rely on butyrate as a principal source of energy, butyrate exposure was shown to increase apoptosis, senescence and differentiation of colon cancer cells lines but promoted cancer development in colon cancer animal models[73–75]. More research is needed to untangle these conflicting effects of butyrate on cancer progression especially in non-intestinal cancers. Butyrate is also known to exert anti-inflammatory activities by affecting a number of

cellular processes including immune cell migration and cytokine expression. However some studies suggest that the immune modulatory activities of butyrate are mainly through the induction of Foxp3+ T regulatory cells (Treg) and the prevention of dendritic cell (DC) maturation[76,77]. A recent study found that high serum butyrate levels produced by gut microbes were associated with resistance to anti-CTLA-4 blockage and a higher proportion of Treg cells in patients with metastatic melanoma[46]. The authors suggested that down modulating butyrate signaling pathway may be mandatory to restore anti-CTLA4 efficacy in patients with high butyrate and other SCFA serum levels. Our results showing reduced fecal butyrate levels after a short 7 weeks treatment with MAG-EPA supplement may represent a promising and simple opportunity to improve the clinical efficacy of anti-CTLA-4 treatments. While immunosuppression may be beneficial to control inflammatory diseases, it has detrimental effects in cancers. PCa micro-environment is mostly immune-suppressive and several studies have shown an association with high density intra-tumoural Foxp3+ Tregs and lethal PCa[76,77]. Lowering gut-produced butyrate levels could also potentially modulate the PCa immunosuppressive micro-environment to release an effective anti-tumoural immune response representing yet another mechanism of its activity. Accordingly, more studies are needed to identify the cellular targets of microbiota-derived SCFAs in PCa patients.

Overall, our studies revealed the existence of a crosstalk between PCa growth and the host gut microbes particularly those involved in the metabolism of LCFA. Supplementation with LCn3 resulted in a reduction of cancer growth in mouse models and reduction of upgrading in pre-prostatectomy patients associated with a reduction of gut Ruminococcaceae and fecal butyrate levels. It identifies one potential mechanism to explain the beneficial effect of diet rich in LCn3 in reducing the initial diagnosis and progression of PCa. This conceptual advance is important since targeting the microbiota via the diet is non-invasive, applicable to most cancer patients and could limit cancer progression in stages of the diseases where patients are not receiving other treatments.

## Methods

### Human samples

This research complies with all relevant ethical regulations, the study protocol was approved by the Centre Hospitalier Universitaire (CHU) de Québec-Université Laval's review board (#2012-1012) and the trial registered to clinicaltrials.gov (NCT02333435). Written informed consent were received for all clinical data and samples and was described in the published parental clinical trial[21]. A panel of 62 prostate cancer (PCa) patients with Gleason Grade Group ≥2 localized PCa agreed to provide fecal sample $5.0 \pm 0.5$ weeks (mean ± SEM) on average before they were treated by radical prostatectomy (RP) and on the day before the surgery. Briefly, the International Society of Urological Pathology (ISUP) agreed on a system of five prostate cancer Grade-Groups based on the proportion of the three Gleason patterns 3, 4 or 5[39–41], which are associated with prostate cancer progression risk. Grade group 1 tumours with only Gleason pattern 3 are indolent and most patients are recommended active surveillance. Grade Groups 2, 3 and 4 are defined by the proportion of pattern 4 (<50%, >50% and 100% respectively), and Grade Group 5 have pattern 5, which is the most aggressive cancer. We also collected fecal samples from 47 PCa patients that had their prostate surgically removed by RP on average $5.2 +/−1.0$ years (mean ± SEM) before sample collection. These patients experienced different levels of cancer recurrence based on Prostate-Specific Antigen (PSA) serum levels; low PSA values (between 0.05−0.49 ng/mL) and 18 patients had clinically significantly high PSA values (above 0.50 ng/mL). Another set of 41 human fecal samples were obtained from a subgroup of men participating into a phase IIb double-blind randomized controlled trial testing 3 g/day of MAG-EPA versus placebo for 7.2 weeks (±0.37) on average before their radical

prostatectomy[21,37]. All patients in the pre-prostatectomy group (Fig. 1a–d) received one dose of 500 mg Ciprofloxacin on the day of the prostate biopsy which was performed on average $22.8 \pm 1.43$ weeks prior to sample collection in all pre-prostatectomy patients. While antibiotics may affect the microbiota, all patients had a similar exposure. As for patients with or without PSA failure after radical prostatectomy (Fig. 1e–h), all patients received one dose of intravenous Cefazolin (2 g) at the beginning of surgery. All fecal samples from this cohort were collected several years after surgery. This clinical study was approved by our IRB from CHU de Québec–Université Laval (2012-1012) and registered to clinicaltrials.gov (NCT02333435) and recruited between February 2015 and June 2017. Clinical characteristics such as PSA and tumour volume (estimated using pathological report data) were extracted from patients' chart. Tumour volume (in grams) was calculated by multiplying the percentage of the prostate covered by tumour tissue (evaluated by the pathologist) with the prostate mass documented (in grams) in the radical prostatectomy report. Herein, tumour burden refers to the estimated tumour volume and was assessed as low/medium/high based on tertiles. PCa grade at the study enrolment biopsy (clinical grade) and radical prostatectomy (pathological grade) was assessed using our usual institutional pathological procedures for PCa. Microbiota analyses were conducted on a subgroup of 41 patients that also consented to provide clinical data and fecal samples for microbiota high-throughput analysis (2012-671, 2017–3231). Patients were not involved in the design, conduct or reporting of our research and all were men. Stool samples were collected according to Human Microbiome Project protocol and patients were compensated 20 CAD for providing their fecal samples. The stool kit contained a primary container for a full bowel specimen and single tubes for aliquoting the stool sample. Patients were instructed to place the stool collection frame on the back of the toilet bowl, then to place the collection bowl in the frame and were specifically instructed to avoid any contact between the stool sample and urine. Stools were excreted directly into the 650 mL commode specimen collection system (Fisher scientific Company, ON, Canada). Subsequently, dedicated tubes (Norgen Biotek, Ontario, Canada) were used to scoop the stool sample into separate aliquots samples that will be used for DNA sequencing and stored in the biobank. Stool samples were immediately frozen and conserved at −20 °C and then at −80 °C for biobanking purposes.

### 16S rRNA metataxonomic analysis

For bacterial DNA extraction, fecal samples were homogenized using a minilys bead-beater apparatus (Bertin, Rockville, MD). Samples were lysozyme and proteinase-treated before phenol-chloroform extraction and ethanol precipitation. Fecal DNA samples were further purified using DNeasy Blood and Tissue Kit (QIAGEN, Missisauga, ON, Canada) before 16S library amplification. Libraries were amplified as described before[78]. Briefly, for each fecal DNA sample, a fragment of the V3-V4 hypervariable region of the bacterial 16S rRNA gene was amplified by PCR with bar-coded primer. High-throughput sequencing of the bar-coded amplicons was performed on a MiSeq apparatus (Illumina, San Diego, CA) by the Institut de Biologie Intégrative et des Systèmes (IBIS-Université Laval, QC, Canada) services. Bioinformatic analysis of the reads was described before[78] and began with the removal of low-quality reads with Mothur. Next, sequences were filtered based on the abnormal presence of uncertain bases (N), invalid DNA length, sequences that began with incorrect primer or that contained series of height homopolymers or more. Sequencing primers were trimmed out and Mothur was used to identify and eliminate chimeric reads and sequences considered contaminants. Remaining sequences were then aligned against SILVA reference alignments using the k size = nine parameters. To allow comparison, reads were normalized using the sample with the lowest total read number for rarefaction. Sequences that successfully passed the previous filtering steps were used in

Mothur to generate a pairwise distance matrix to allow reads to cluster into OTUs (Operational Taxonomic Units) using the furthest-neighbor algorithm and specifying a distance cutoff value of 0.03. OTUs occurring from a single read were discarded at this step. Taxonomical classification of OTUs was carried out in Mothur with reference sequences based on the Greengenes database. The sequence identity threshold of 97% allows reasonable discrimination of sequences at the genus-level. The methods used to analyse the bacterial taxonomic profiles of the murine gut microbiota are described in[78]. All samples were rarified to the same sampling depth of 3676 reads. Before quality check, a mean of 65,720 raw reads were generated for each sample, whereas after merging, quality filtering, and mapping steps, the average number of reads per sample was 14,854. Alpha diversity indexes were calculated using PAleonto-logical STatistics software (PAST Version 3.18). For beta diversity analysis, the microbiota reads were analysed using MicrobiomeAnalyst[79]. We employed the Statistical Analysis of Metagenomic Profiles (STAMP v2.1.3) statistical probability model to identify biologically relevant differences between metagenomic communities[80]. Phylogenetic Investigation of Communities by Reconstruction of Unobserved States (PICRUSt)[30] was used to extrapolate functions associated with 16S rRNA sequencing data. STAMP was used to generate Principal Component Analysis (PCA) and to identify differentially abundant taxa and predicted functions determined by PICRUSt between groups. The datasets generated during/or analysed in the current study will be deposited in the sequence read archive from the national center for biotechnology information.

## Murine models and treatments

**TRAMP-C2 model.** Immunocompetent C57BL/6 N male mice (Charles River 027) were injected with $2.0 \times 10^6$ TRAMP-C2 mouse PCa cells (subcutaneous) to measure changes in the gut microbiota associated with tumour growth. The TRAMP-C2 study protocol was approved by our Institutional Review Board (IRB) of CHU de Québec – Université Laval, Canada (#2015112). For the TRAMP-C2 model, twelve 6–8 weeks old C57BL/6 mice per group were fed with a low-fat diet (#D12450H, Research Diet Inc., New Brunswick, NJ) ad libitum and animals were single-housed throughout the experiment and under a 12 h day-night cycle with lights off at 18:00. Relative humidity was maintained between 40% and 60% and room temperature between 20 and 22 °C. 0.2% of the total fat in the low-fat diet was arachidonic acid (AA) but the diet contained no DHA or EPA. To evaluate the effect of specific fatty acids on TRAMP-C2 tumour growth, mice were daily supplemented by oral gavage with 618 mg/kg of body weight of purified MAG-EPA (SCF Pharma, Ste-Luce, Canada), MAG-DHA (SCF Pharma), MAG-AA (Nu-Check-Prep Inc, Elysian, MN) or HOSO (SCF Pharma) before tumour cell injection and until animal sacrifice. The dose supplemented mimicked the clinical dose used in the clinical trial (3 g/day of MAG-EPA, NCT02333435). Tumour size was measured every other day and the tumour volume was calculated using the following formula: (4/3) * 3.14159 * (longest diameter/2) * (shortest diameter/2) ^ 2. Tumour growth end point was reached at 2 cm³ total tumour volume (when both flanks were injected, end point is sum of both tumours). Tumours were collected from each mouse at sacrifice and stored at −80 °C. Fresh feces were collected throughout the study and immediately stored at −80 °C. For Fecal Microbiota Transplantation (FMT) experiments, C57BL/6 N 6–8 weeks old male mice, fed ad libitum chow diet (2018SX, Inotiv) under a 12 h day-night cycle with lights off at 18:00, were treated with 1 mg/mL of ampicilin in the drinking water that was changed every 2–3 days. In addition to the drinking antibiotic, mice were gavaged with 5 mg/mL of Streptomycin and 1 mg/mL of Colistin daily. Antibiotic treatments lasted for 8 consecutive days and animals had 48hrs to washout the antibiotics before being gavaged fecal samples from PCa patients or control. Control mice received their pre-antibiotic tumour-free PBS-resuspended feces. Receiver mice were

given the PBS-resuspended feces from six independent PCa patients by intragastric gavage. Prior to the intragastric gavage into recipient mice, human fecal samples were culture tested for viability and FMTs were administered to the animals twice for two successive weeks and every time a drop of fecal suspension was added to the animal fur. Again, $2.0 \times 10^6$ TRAMP-C2 cells were injected into both flanks of animals after their last FMT. Tumour size was measured every other day and the tumour volume was calculated as described above.

**Pten$^{-/-}$; Rb1$^{+/+}$ and Pten$^{-/-}$; Rb1$^{-/-}$ models.** Experiments were conducted in the Animal Resources Facility at the Research Institute of the McGill University Health Centre (RI-MUHC). The animal protocol followed the ethical guidelines of the Canadian Council on Animal Care, and was approved by the RI-MUHC Glen Facility Animal Care Committee. Three weeks-old immunocompetent male C57BL/6 J mice (The Jackson Laboratories 000664) were fed a purified diet (TD.130838, Envigo) ad libitum under a 12 h day-night cycle with lights off at 18:00. Animals were injected subcutaneously with 50/50 mix of $1.5 \times 10^6$ Pten$^{-/-}$; Rb1$^{+/+}$ or Pten$^{-/-}$; Rb1$^{-/-}$ murine PCa cells[26,27] and phenol red-free Matrigel (CB-40234C, Corning) at 9 weeks of age. At about 13 weeks of age, tumour volume was recorded and feces from both tumour models and matched tumour-free animals were collected and stored at −80 °C ($n = 8$/group).

## qPCR

Fecal DNA extracted from feces was tested by qPCR for the differential abundance of bacteria using the delta-delta Ct approach. Primers were ordered from Integrated DNA Technologies (IDT Company, Coralville, IA) and sequences were; Ruminococcaceae (for- actgagaggttgaacggcca, rev-cctttacacccagtaawtccgga), Parabacteroides distasonis (for-gaacgc-tagcgacaggctta, rev- gatccctgcttcatgcggta), Lactobacillus ruminus (for-aagaccgcgaggtttagcaa, rev-tagacggctggctccaaaag) and control total 16S rRNA (for-tggagcatgtggtttaattcga, rev-tgcgggacttaacccaac). Catalase (for-tttYaaYcgagaRMgRgtNccNgaRMgNg, rev-gaatatgcaaaaaggcgNcc YtgNaRNaKYttRtc). Butyrate kinase (for- tgctgtWgttggWagaggYgga, rev- gcaacIgcYttttgatttaatgcatgg). Butyryl CoA transferase (for- gacaa gggccgtcaggtcta, rev- ggacaggcagatRaagctcttgc). The qPCR reactions were prepared using the Advanced qPCR mastermix with Supergreen Low-ROX (Wisent Inc., St-Bruno, QC), 2 units of FastStart Taq DNA polymerase, 500 µM of primers and 2 µl of genomic DNA in a total reaction volume of 20 µl. For the human fecal samples, a DNA-free Taq was used (cat#101005, Boca scientific) to amplify specific bacteria-related genes (Catalase, Butyrate kinase and Butyryl CoA transferase). The cycling conditions consisted of 1 cycle at 95 °C during 5 min for denaturation and hot start, followed by 40 three-segment cycles for amplification (94 °C for 30 s; 55–60 °C for 45 sec; 72 °C for 30 s). Melting curve was performed at the end of PCR amplification to verify the reaction specificity. Experiments were run on StepOneplus Real Time PCR System (ThermoFisher Scientific, Waltham, MA).

## Short-chain fatty acid analysis

For short-chain fatty acid analysis, fresh frozen human fecal samples were resuspended at 100 mg/mL in water and later homogenized using a beadruptor apparatus (Omni international, Kennesaw, GA). After homogenization, fecal samples were low speed-centrifugated and the supernatant was used for acetic acid extraction. Purified samples were injected in a gaz chromatography-flame ionization detector along with standards for quantification. Samples were extracted and measured in duplicate and the mean of both separate SCFA quantifications were used for statistical analysis.

## Statistical analyses

Difference of tumour growth between groups was performed by two-sided student's t-test, two-sided Welch's t-test, Wilcoxon test and other statistical analyses are described in the figure legends. Unless specified

otherwise, the data is presented in the text as mean ± SEM. *P*-values below or equal to 0.05 were considered significant. NS or no asterisk = not significant, $p \geq 0.05$, *$p \geq 0.05$, **$p \geq 0.01$, ***$p \geq 0.001$.

### Reporting summary

Further information on research design is available in the Nature Portfolio Reporting Summary linked to this article.

## Data availability

All tools used for 16S metagenomic are available online; Greengenes [https://mothur.org/wiki/Greengenes-formatted_databases], RDP reference file [https://mothur.s3.us-east-2.amazonaws.com/wiki/trainset9_032012.pds.zip], SILVA reference alignments [https://mothur.org/wiki/Silva_reference_files] and Mothur [https://mothur.org/].The raw sequence reads generated in this study have been deposited in the NCBI Sequence Read Archive and are directly available under accession code PRJNA1047087. Source data are provided with this paper. Other information is available in supplementary information and otherwise available upon request. Source data are provided with this paper.

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

## Acknowledgements

The authors first thank the participating men who made this work possible and the *Banque de tissus et de données médicales pour la recherche sur les maladies prostatiques de l'Hôtel-Dieu de Québec* biobank for providing feces samples and clinical data. The authors thank Dr. Samuel Fortin from SCF Pharma for having kindly provided MAG-EPA, MAG-DHA and placebo HOSO capsules for this clinical trial and for the mouse work. Authors also thank Hélène Hovington for clinical databases, Pierre-Luc Mercier for DNA extraction, Marie-Josée Bilodeau for patient recruitment, Perrine Feutry for SCFA analyses, IBIS team for all the next-generation sequencing, Dr. Aurélie Le Page, Serena Bou Dagher and Clara Blancfuney for technical assistance regarding the mouse work as well as Xavier-Philippe Légaré for revisions to the manuscript. This research was funded by the Canadian Cancer Society and Movember Foundation (grant #400345/D2016-1336) to V.F., N.D. and A.M., by the W. Garfield Weston Foundation through Weston Family Microbiome Initiative awarded to V.F., Y.F., F.R and A.M., and a Canadian Institutes of Health Research project grant awarded to C.A.P. (PJT-148821) and to D.P.L. (PJT-162246). A.M. was supported by a CIHR/Pfizer research Chair on the pathogenesis of insulin resistance and cardiometabolic complications. G.L. was supported by Diabetes Canada post-doctoral fellowship (NOD_PF-3-15-4764). J.L. was supported by the Fonds de Recherche du Québec - Santé (FRQS) scholarship. N.G. was supported by PCa Canada Ph.D. scholarship. T.H. is a 100 Days Across Canada Bursary Award Recipient for PCa research. I.B.J. and F.R. are associated with the Canada excellence research chair on the microbiome-endocannabinoidome one axis in metabolic health. D.P.L. is a William Dawson Scholar of McGill University, a Lewis Katz – Young Investigator of the Prostate Cancer Foundation and is a Research Scholar – Junior 2 from the (FRQS). V.F. is a recipient of FRQS clinician-scientist career award.

## Author contributions

G.L., N.G. and F.G. contributed to the design and laboratory experiments. A.F., T.H., N.B., C.A. and D.P.L. contributed to the design and laboratory experiments of the Pten/Rb1 PCa mouse models. L.E. provided the Pten−/−; Rb1+/+ and Pten−/−; Rb1−/− PCa cells lines. G.L., K.R., and J-F.P. collected clinical samples and data. H.W.X., S.B., N.D. and T.V. performed the bioinformatics and biostatistical analysis of the data. G.L., K.R, T.V., I.B.J., H.W.X., S.B., N.D., A.B., F.R., Y.F., D.P.L., A.M., and V.F. contributed to figure preparation, data interpretation and writing of the manuscript. All authors approved the final manuscript.

## Competing interests

The authors declare no competing interests.
