## [Peer Review File · Nature Communications]

Reviewers' expertise:

Reviewer #1. Microbiome and cancer

Reviewer #2. Microbiome and prostate cancer

Reviewer #3. Prostate cancer and mouse models

REVIEWER COMMENTS

Reviewer #1 (Remarks to the Author):

NCOMMS-22-05203

"The gut microbiome-prostate cancer crosstalk is modulated by dietary polyunsaturated long-chain fatty acids" by Lachance and co-workers.

In recent years, it has become clear that changes in the gut microbiota are involved in the tumourigenesis of many cancers, not only colorectal cancer. In this paper, Lachance and co-workers attempted to explore the role and mechanisms of gut bacteria in increasing tumour burden in prostate cancer. Using clinical samples and mouse models, the authors found that faecal microbiota transplantation (FMT) from patients with high tumour volume promotes prostate cancer growth in mouse models. Furthermore, faecal metabolic pathway analysis of the mouse model revealed that long-chain fatty acids (LCFAs) are involved in prostate cancer growth. Dietary supplementation with LCFA omega-3 inhibited tumor growth in mouse models and malignant progression at prostatectomy in prostate cancer patients, and was associated with lower fecal levels of Ruminococcaceae in both. Analysis of metabolites produced by bacteria of the Ruminococcaceae family in faecal samples from patients treated with Omega-3 supplements showed a selective and significant reduction in butyrate, a gut bacterial metabolite. From these results, the authors concluded that butyrate plays an important role as one of the mechanisms by which a diet rich in omega-3 fatty acids suppresses prostate cancer.

This is a potentially interesting paper as it is based not only on data from mouse models, but also on human clinical data, and is expected to be developed into clinical applications. However, there are several limitations, which in my view preclude its publication in Nature Communications, at least in its present form.

Critiques:

1) In this paper, the authors emphasize only the pro-tumourigenic effects of Butyrate. However, importantly, butyrate reportedly possesses both tumour promotive and suppressive properties through energetic or epigenetic functions, depending on the cell type, duration, and amount of exposure (see for example but not limited to PMID: 24766895; PMID: 25198138). Therefore, the authors should be cautious about this point.

2) The authors emphasize that reduced intestinal butyrate levels may inhibit prostate cancer progression via reduced Tregs and other anti-immune effects, but does butyrate really play an important role in promoting prostate cancer progression? No experimental evidence has been presented as to whether the action of butyrate is really mediated by Tregs or other anti-immunity effects. I feel there needs to be some experimental evidence to support the authors' hypothesis.

3) Along the same lines, butyrate may promote tumorigenesis by enhancing Ttrg production, but may also promote tumorigenesis by inducing senescence-associated secretory phenotypes (SASP) (PMID: 34584098; PMID: 22422937). Therefore, the authors should investigate whether MAG-EPA administration reduces Tregs and/or senescent cells in prostate cancer tissue, at least in the mouse model.

4) The authors have examined α -diversity of the gut microbiota throughout this paper, but have never examined β -diversity. The authors should also examine β -diversity.

Reviewer #2 (Remarks to the Author):

This is a very interesting study and a substantial piece of work. I have some questions and suggestions, which I feel could improve clarity and should be addressed before the manuscript may be accepted for publication.

1. The description of human samples/patients cohorts in the methods section lacks more details. Where/when were the patients recruited and samples collected (hospital, date/year)? Basic clinic

information for all individual patients is missing, and should be consistently reported for both patient sets so that they can be more clearly compared (age, ethnicity, PSA level at diagnosis/treatment, TNM stage and Gleason Grade at diagnosis and at RP).

2. Have any of the patients taken antibiotics – and when, as compared to time of fecal sampling? Most often antibiotics is used prior to prostate biopsy, and this could affect the gut microbiome?

3. The quality of the 16sRNA seq data is difficult to assess. Not clear how many reads were generated for each sample (total/mapped), before QC and after QC (both human and mouse data).

4. I do not feel, that the results in figure 1 are convincing. There seems to not be a consistent pattern in the two patient sets for neither Firmicutes or Bacteroidetes abundance, nor for alpha diversity. Another major limitation in this part is that it is underpowered, with very small numbers in some of the groups, which is particularly problematic given the significant heterogeneity (wide error bars) and the inconsistency between the two patient sets.

The PCA plots in Suppl Fig 1 also seems to not shown any clearly distinct separation between the 3 subgroups defined for each patient set. The two patient sets also seem to be rather different, with samples collected prior to RP versus at BCR (where the whole prostate has been removed) – or at least that is how I understand it?

5. Figure 1A and 1E are not very informative, would be useful with more clinical information about these patient sample sets. See my comment no 1.

6. There are samples from 8 mice in figure 2B/E, but 12 mice in figure 2A/2D. How come? Any selection?

7. The data in figure 2B (mouse data) seems to conflict with the observations in fig 1 (patient data). In patients, the authors report a trend toward higher abundance of Firmicutes and lower abundance of Bacteroidetes in patients with higher PSA/tumor burden, but the opposite pattern is seen in the mice (higher tumor burden associated with lower Firmicutes and higher Bacteroidetes). In fact, there seems to be a pattern through the whole manuscript, that only a selected part of the results presented are consistent with the overall conclusions made and final “biological model” presented.

8. How do the authors explain the opposite direction of changes in abundance of Firmicutes in the different mouse models (Fig 2B vs 2E), and the difference in the PCA plots in the different mouse models (compare Fig 2C and 2F).

9. What is the Gleason score for each donor (Fig 3C). How were the donors selected? Were their microbiota composition known beforehand (16S rRNA seq data)?

10. Add dots for individual mice in figure 3D, so easier for readers to assess variability. Error bars show SEM, which will visually minimize variation.

11. It would be interesting to compare the results in Figure 4A+B, by performing similar analyses using the data for the two patient data sets shown in figure 1/5. Is the PTEN/RB1 status known for the patient tumors?

12. The results shown in figures 5D-F are difficult to interpret. How can you tell the cause of down/upgrading from biopsy to RP? Systematic TRUS biopsy is notoriously error prone due to subsampling errors, leading to re-classification of say >40% of PC tumors at radical prostatectomy. Hence, it would be crucial to know if the clinical characteristics of the patients in the two groups (HOSO vs. MAG-EPA) are comparable – if not, it may not be surprising that they differ in the proportion of up/down-graded cases.

13. In continuation of this, do the authors suggest that treatment with MAG-EPA over a period of only 7 weeks can alter the gut microbiota (Fig 5F indicates that this is rather heterogeneous) and in turn that this will alter the Gleason grade of the PCa tumor (Fig. 5E)? Or is it perhaps more likely, that the reclassification is mainly (or exclusively) due to the error-prone biopsy procedure (misclassification of Gleason score). Would be interesting to know what the change in the microbiome is over time in patients not receiving a supplement (negative controls)

14. Furthermore, it would be interesting to know if the relative abundance of Ruminococcaceae in samples at baseline and at RP, respectively, correlate with the true Gleason Grading for that patient (i.e. determined at RP, not at TRUSbx).

15. Matched data is available for the results presented in figure 6A-F. Would be helpful to also show those rather than just the averages, to better understand the possible heterogeneity in the effects. What is the rationale for not including a no-treatment group as a control – could reveal the level of variation/heterogeneity in signals?

16. I do not feel the compared groups in figure 6G are meaningful, for the same reasons as stated in my comments 12-13.

Reviewer #3 (Remarks to the Author):

General Comments:

1. The study of gut microbiota is essential for gaining a more thorough understanding of the interactions of the microbiota with oncogenesis, tumor progression and cancer therapy. Therefore, this study provides new knowledge on the potential role of the status of retinoblastoma (Rb) gene expression and the gut microbiota on prostate cancer.

2. Figures 4 and 5 are perhaps the most interesting in the entire study. The use of the three mouse models, the in-depth analysis of differences between their microbiota, the implication of loss of Rb expression and/or function in altering microbiota and how this could modulate tumor growth and

the response of tumor growth to treatment with different long-chain polyunsaturated fatty acids would make a very interesting and insightful manuscript on its own.

3. The inclusion of Figure 1 in this study is unclear. While an in-depth analysis of these human fecal samples has great potential, this was not realized in the overall experimental design of the manuscript. Furthermore, there is a concern that the formula used to determine low, medium and high tumor burden might have skewed the data in this study (as discussed below). In addition, the link between the data in Figure 1 and the remainder of the paper is tenuous. Even the bacteria analyzed (Firmicutes and Bacteroidetes) appear to be of minor importance in the mouse models where Parabacteroidetes, Lactobacillus, and Ruminococcaceae take center stage. This is confirmed in the experimental design where (a) Rb status in mouse tumors was known; Rb status in human tumors was not known, (b) mouse fecal samples were thoroughly analyzed using multiple experimental and analytical approaches (figures 3, 4, SF2, SF3 part1, SF3 part2, SF4, and 5); human fecal samples were not, etc. The only exception appears to be the clinical study where MAG-EPA PUFA or HOSO was administered to patients prior to prostatectomy (Figures 5D-5F, S5) and human fecal material before and after prostatectomy were analyzed. Therefore, it might be advantageous to consider the value of the human fecal samples collected in Figure 1 and design a comprehensive study that would optimize their value.

4. Overall, the Results Section tends to be descriptive. For example, in Figure 2, descriptions include, “showed a modulation”, “were distinct”, and “had similar fecal microbiota”. In Figure 5, examples include “... there was an up-grading toward a more malignant cancer....., a down-grading toward a less malignant cancer....” and “significantly reduction”. The term grading was frequently used, but grading scores were never provided, making it difficult to interpret the data. This study would be considerably strengthened by the use of analytical scientific language, by providing a rationale where appropriate and by succinct and quantitative analyses (in addition to p values) that support the interpretation of the results.

Other Comments:

1. Materials and Methods, Human samples. Please provide additional details for the different patient cohorts/groups investigated in this study, for example:

Gleason Grade Group ≥ 2 localized PCa cohort:

a. Please define Gleason Grade Group ≥ 2 localized PCa as researchers outside the field may not understand the prostate cancer classification scale.

- b. Please provide the number of patients providing fecal samples and range of weeks prior to prostatectomy
- c. Provide the number of patients providing fecal samples and range of weeks after prostatectomy
- d. Further subdividing these samples into additional categories would help with the interpretation of the data in Figure 1. Examples include:
- Gleason Score
 - defining the range of PSA levels for each category which are currently only described as “low, medium, and high”
 - the range of % tumor verses benign areas in the tissue samples.
 - Tumor volume – approximation as determined by the area under the tumor margins inked in by the pathologist
 - other variables considered important to the interpretation of the data, e.g., age (as prostate volumes tend to increase with age and BPH content)
- e. Were the patients who provided fecal samples before radical prostatectomy the same group of men who also provided fecal samples after prostatectomy? If yes, could the “before” and “after” fecal samples from the same patient be compared? This linear analysis would increase statistical power considerably.
- f. Was the range of fecal collection time prior to prostatectomy the same for the two groups described above as that for men who participated in the phase IIb/MAG-EPA trial, i.e., 5.2 weeks (\pm 4 days)?

Phase IIb double-blind randomized controlled trial group:

In a similar manner, please define the samples and other pertinent information for the patients who provided fecal samples for this current study.

Subgroup of 41 patients

In a similar manner, please define the samples and other pertinent information for the subgroup of 41 patients who consented to provide clinical data and fecal samples for microbiota high-throughput analysis:

- Including the additional 24 patients with biochemical recurrence.
- including the 3 patients with undetectable PSA levels

Using a table might be helpful in describing the different groups.

2. Materials and Methods. Please provide a section that briefly, but succinctly, describes how the samples for Figures 1E through 1H were analyzed. This study uses the term “tumor burden” to describe the % cancer tissue / total prostate weight at surgery (g). “Tumor burden” is a clinical term that is more frequently used to describe the degree of metastasis in the body. Therefore, please provide a clear description of how the term is used in this study to prevent confusion. Alternatively, a term that pertains to the samples being analyzed, perhaps tumor tissue (volume) or tumor tissue (%) could be used.

3. Materials and Methods, Murine models and treatments. Regarding the experimental design:

a. For the TRAMP-2C study, please indicate that the cells were injected s.c. Similarly, please clarify how Pten^{-/-}; Rb1^{+/+} and Pten^{-/-}; Rb1^{-/-} cells were administered. Currently the text states that these cells were “injected in the flank”. This description could mean s.c., or it could also mean intramuscular.

b. Regarding the protocol for generating the tumors, please clarify whether (a) tumors were established first and then mice were treated or (b) tumor cells were injected after (or along) with treatment at the same time. The first approach would determine effect of treatment on established tumors. The second approach would determine the effect of treatment on the success of establishing tumors in addition to whether treatment would promote tumor growth (or regression).

c. Mice were fed with a low-fat diet. Did this diet contain any of the specific fatty acids of interest?

d. Mice were given 618 mg/kg of body weight of purified MAG-EPA daily via oral gavage. Does the mouse dose mimic the dose (3g/day of MAG-EPA) given to men participating in the Phase IIb clinical trial?

e. For the FMT study, please indicate whether oral or intragastric gavage was used. If oral gavage was used, would this approach cause any alterations in bacterial survival rates and/or gene expression profiles as the donor microbiota pass through the environment of the stomach?

f. Please provide a brief description of how the FMT material was prepared for gavage, what steps were taken to ensure a good representation of both aerobic and anaerobic bacteria in the specimens, and the administration regime used to maintain the human donor microbiota in recipient mice.

g. Was the overall bacterial load in every human donor microbiota similar? If some samples had a low microbiota load while others had a high microbiota load, would this impact the results?

h. Please indicate the sex of the recipient mice. Males would provide the optimal supportive testosterone microenvironment for prostate tumor growth.

4. Results section, Figure 1A-1D. These panels compare differences in PSA and bacterial content of samples obtained from patients showing biochemical resistance (BCR). This analysis relies primarily on PSA levels. While increasing PSA levels remain useful as a biomarker for BCR, higher Gleason scores with lower PSA levels have also been observed in patients who are newly diagnosed with aggressive prostate cancer. Did the study criteria include MRI scans? If yes, then the scans would confirm biochemical recurrence as determined by increased lesion size and/or metastatic spread (i.e., increased tumor burden).

5. Results, Figure 1E-1H. Figure 1E investigates tumor burden as defined by the % of cancer tissue / total prostate mass (g) at surgery. This analysis assumes that any changes in the weight of the total prostate correlates directly to the weight of a tumor in the prostate. However, the total weight of each patient prostate will be different; and total prostate weight will likely increase with age and contain differing amounts of BPH. Therefore, including the variables of weight, age and BPH content into the calculation could potentially “mask” the levels of cancer tissue in the whole prostate. Comparing % cancer area (and cancer stage?) versus benign area per biopsy (where n = the number of tissue slides analyzed) to total tumor volume as defined by the total area under the positive tumor margins inked on the biopsy tissue slices/pieces by the pathologist might be more informative and statistically significant. Similar comments pertain to the X axis in the analysis of bacterial content in 1F through 1H.

6. Results, Figure 2. Fig. 2A lacks a quantitative description of the growth curve. In analyzing the growth curve, it appears to be a classic sigmoidal growth curve where tumor volume increases slowly but steadily up to ~450, and then the slope of the line changes and tumor volume increases exponentially from ~500 to just below 2000 after which it plateaus at ~2000. When determining where to establish the exponential growth group, it is unclear as to why this group was selected at the point where exponential growth is initiated (~500). Please provide a rationale for the selection of this tumor volume.

In Fig.2B, the conclusion stated that only Bacteroidetes showed a statistical change in abundance. Please note the changes in Firmicutes abundance during exponential growth and endpoint stages. Are these changes not statistically significant? If not, how do these observations relate to those observed in human fecal material in Figure 1?

The experimental approach and analysis of the data in panels 2C and 2F are limited. Using panel 2C as example, is 2C a bicompositional biplot? What samples do the left and right panels represent? Clearly the samples on the left panel show more separation where the “exponential growth” and “end point” groups cluster together, whereas the samples on the right panel show significant overlap.

The rationale for the inclusion of Pten^{-/-};Rb1^{+/+} and Pten^{-/-};Rb1^{-/-} PCa cell-derived tumors appeared to be that they were available. Please provide a rationale for including these PCa cell-derived tumors. For example, is there any correlation between the Pten^{-/-};Rb1^{+/+} and Pten^{-/-}; Rb1^{-/-} and TRAMP-2 tumors and human PCa (e.g., genotype; histopathology; etiology, etc.), that may have modulated the impact of hFMT on tumor growth? This information would be useful at this point in the manuscript, and not just in the Discussion. In addition, the protocols on when tumor cells were injected and when treatments were given were not always clear (please see Comment 3b).

7. Results section, Figure 3. Please consider Comments 3d through 3f when describing the samples, data, analysis and interpretation of the data.

In Panel 3C and D4, the authors conclude that, “Interestingly, human PCa FMT2 and FMT3, corresponding to the patients with high tumor volume at radical prostatectomy, but not FMT1 corresponding to a patient with low tumor burden (Fig.3C), significantly increased TRAMP-C2 tumor burden of the recipient animals compared to the control mice at the experimental endpoint (Fig.3D).” While a comparison of the three hFMTs to the mFMT control suggests that hFMT2 and hFMT3 cluster together and are statistically significant from hFMT1, a comparison of the three hFMTs to each other would probably indicate that given the considerable overlap of the error bars, all three hFMTs stimulated TRAMP-C2 tumor growth to a similar degree,.

In the event that all three hFMTs stimulate TRAMP-C2 tumor growth to a similar degree, this might suggest that the degree of tumor burden has little effect. For consideration is that the formula used for calculating tumor burden could have impacted the analysis of these data.

Alternatively, there is a phenomenon called “super-donors” where one donor’s fecal material results in significantly more successful FMT outcomes than that of other donors. Clearly, additional samples would need to be tested to confirm any results since only 1 low tumor burden and 2 high tumor burden-derived FMTs have been tested.

In the Figure 3 legend, please define the acronym “J4”.

8. Results section, Figures 4 and 5. In general, p values indicate that a change is statistically significant within the parameters of the experimental design; however, they do not describe what the change is. As indicated above, these data are basically descriptive, and changes (increases or decreases) are never quantified. Of note is that the y axes in figures 4C and 4D are an order of magnitude different when analyzing Lactobacillus 16S rRNA in fecal samples from mice with TRAMP-C2 tumors compared to fecal samples in mice without tumor, with Pten^{-/-}; Rb1^{+/+} or Pten^{-/-}; Rb1^{-/-}, indicating that

Lactobacillus load in fecal samples from the transgenic models is considerably greater than that from the TRAMP-C2 model. In contrast, Parabacteroidetes load is greater (albeit more modestly) in fecal samples from the TRAMP-C2 model as compared to the transgenic mice. Therefore, while the results indicate that there are common gut microbiome alterations, the degree of these alterations might be tumor specific.

NCOMMS-22-05203

Rebuttal, point-by-point reply to the reviewers

Reviewer #1

NCOMMS-22-05203

"The gut microbiome-prostate cancer crosstalk is modulated by dietary polyunsaturated long-chain fatty acids" by Lachance and co-workers.

In recent years, it has become clear that changes in the gut microbiota are involved in the tumourigenesis of many cancers, not only colorectal cancer. In this paper, Lachance and co-workers attempted to explore the role and mechanisms of gut bacteria in increasing tumour burden in prostate cancer. Using clinical samples and mouse models, the authors found that faecal microbiota transplantation (FMT) from patients with high tumour volume promotes prostate cancer growth in mouse models. Furthermore, faecal metabolic pathway analysis of the mouse model revealed that long-chain fatty acids (LCFAs) are involved in prostate cancer growth. Dietary supplementation with LCFA omega-3 inhibited tumor growth in mouse models and malignant progression at prostatectomy in prostate cancer patients, and was associated with lower fecal levels of Ruminococcaceae in both. Analysis of metabolites produced by bacteria of the Ruminococcaceae family in faecal samples from patients treated with Omega-3 supplements showed a selective and significant reduction in butyrate, a gut bacterial metabolite. From these results, the authors concluded that butyrate plays an important role as one of the mechanisms by which a diet rich in omega-3 fatty acids suppresses prostate cancer.

This is a potentially interesting paper as it is based not only on data from mouse models, but also on human clinical data, and is expected to be developed into clinical applications. However, there are several limitations, which in my view preclude its publication in Nature Communications, at least in its present form.

1) In this paper, the authors emphasize only the pro-tumourigenic effects of Butyrate. However, importantly, butyrate reportedly possesses both tumour promotive and suppressive properties through energetic or epigenetic functions, depending on the cell type, duration, and amount of exposure (see for example but not limited to PMID: 24766895; PMID: 25198138). Therefore, the authors should be cautious about this point.

We thank reviewer #1 for these very insightful comments that made us realize how in the initial submission we had emphasized only one of the many potential mechanisms that may explain how butyrate could promote PCa progression other than the induction of Tregs or other anti-immunity

effects. The many potential mechanisms rightfully identified by reviewer #1 were included in the discussion with references (see below) and the immune-mediated potential effects were de-emphasized. These changes are highlighted in blue in the revised text.

Page 14, lines 472-478: “Butyrate can exert both tumour promotive and suppressive effects and these were more extensively studied in colorectal cancer. While gut epithelial cells rely on butyrate as a principal source of energy, butyrate exposure was shown to increase apoptosis, senescence and differentiation of colon cancer cells lines but promoted cancer development in colon cancer animal models¹⁻³. More research is needed to untangle these conflicting effects of butyrate on cancer progression especially in non-intestinal cancers.”.

2) The authors emphasize that reduced intestinal butyrate levels may inhibit prostate cancer progression via reduced Tregs and other anti-immune effects, but does butyrate really play an important role in promoting prostate cancer progression? No experimental evidence has been presented as to whether the action of butyrate is really mediated by Tregs or other anti-immunity effects. I feel there needs to be some experimental evidence to support the authors' hypothesis.

We also agree that experimental data is needed to support a causative effect of butyrate on PCa progression which prompted us to perform additional experiments. To partially address this latter point we performed additional butyrate measurements in patients with and without PSA biochemical recurrence (BCR) (see revised Fig. 6 H). As better explained in the new manuscript, patients with high PSA recurrence after prostatectomy have metastasis that can be detected with new TEP-PSMA nuclear medicine imaging. We found a significant enrichment in the fecal butyrate levels in these patients compared to those with very low PSA BCR or those without BCR.

We included this **new data in (Fig. 6H)** and the following description in the results section of the revised manuscript:

Page 11, lines 351-356: “To further explore the potential association of increase fecal butyrate with PCa aggressiveness we measured the fecal butyric content in the cohort of PCa patients with or without PSA failure after prostatectomy detailed in Fig. 1E-H. Fecal butyric acid concentrations were significantly higher in patients post-prostatectomy with high PSA levels hence likely metastatic (high PSA BCR, $p=0.02$, Welch's t test) corresponding to Fig.6H, supporting that PCa aggressiveness also translate into an altered microbiota metabolism”.

We also reworded the discussion section to clarify that the evidence supporting causal interactions between butyrate and immune cells were tested in other cancers and has yet to be tested vis-a-vis prostate cancer. Finally, we included a sentence to clarify that more experimental evidence is needed in order to test the functional importance of butyrate levels in promoting prostate cancer progression.

Page 15, lines 491-494: “Lowering gut-produced butyrate levels could also potentially modulate the PCa immunosuppressive micro-environment to release an effective anti-tumoural immune response representing yet another mechanism of its activity. Accordingly, more studies are needed to identify the cellular targets of microbiota-derived SCFAs in PCa patients”

Finally, we reworded the summary to better reflect our experimental data on fecal butyrate levels and PCa.

Page 2: Lines 68-70: “Lowering gut butyrate, a known immune modulator, may partly explain the beneficial effect of omega-3 rich diet on PCa incidence and progression” was changed to “This suggests that the beneficial effect of omega-3 rich diet is mediated in part by modulating the crosstalk between gut microbes, their metabolites and PCa”.

3) Along the same lines, butyrate may promote tumorigenesis by enhancing Ttrg production, but may also promote tumorigenesis by inducing senescence-associated secretory phenotypes (SASP) (PMID: 34584098; PMID: 22422937). Therefore, the authors should investigate whether MAG-EPA administration reduces Tregs and/or senescent cells in prostate cancer tissue, at least in the mouse model.

Deciphering the many potential mechanisms by which fecal butyrate may promote PCa aggressiveness and how MAG-EPA may reduce it through direct metabolic effects on the cancer cells or via immune cell modulation or other mechanisms will require significantly more additional work. Although these experiments are important, we respectfully think that they are out of the scope of the current manuscript that focuses on describing a novel crosstalk between PCa and the gut microbiota and how it can be modulated by Omega-3 rich diet.

Page 14, lines 468-478: “Here, we observed reduced fecal butyrate levels in the MAG-EPA-treated PCa patients in association with significantly less cancer upgrading to a more malignant phenotype at prostatectomy. Conversely, we also observed a significant increase in fecal butyrate levels in early metastatic PCa patients post-prostatectomy (High PSA BCR) compared to those without recurrence suggesting an association with increasing PCa aggressiveness. Butyrate can exert both tumour promotive and suppressive effects and these were more extensively studied in colorectal cancer. While gut epithelial cells rely on butyrate as a principal source of energy, butyrate exposure was shown to increase apoptosis, senescence and differentiation of colon cancer cells lines but promoted cancer development in colon cancer animal models¹⁻³. More research is needed to untangle these conflicting effects of butyrate on cancer progression especially in non-intestinal cancers”.

4) The authors have examined α -diversity of the gut microbiota throughout this paper, but have never examined β -diversity. The authors should also examine β -diversity.

As suggested by reviewer #1, we examined β -diversity metrics and provided the **new data** in the revised manuscript for the two patients' cohorts (Fig. 1D and H) and for the two PCa mouse models (Fig. 2 D and H).

Specifically, data from figure 1D and H were tested for β -diversity using the Bray-Curtis index and the data is presented as Principal Coordinates Analysis (PCA) plots in the new Figure 1. We did not observe a significant difference in the beta diversity for the fecal samples related to localized prostate cancer patients in relation to tumour volume (Figure 1D, p-value=0.48, Permanova). However, we noted a significant difference for the beta diversity of fecal samples collected from prostate cancer patients with early metastasis i.e. high PSA BCR (Figure 1H, p-value=0.03, Permanova).

Page 4, lines 142-146: "To test if patients with high tumour volume systematically displayed changes in their microbiota, we measured the beta diversity using the Bray-Curtis index for the whole cohort. Principal coordinates analysis (PCA) did not show difference in beta diversity for those patients (Fig. 1D, p-value=0.48, Permanova)".

Page 5, lines 169-172: "Beta-diversity analysis also showed a significant shift in the high PSA population compared to low PSA or no BCR patients (Fig. 1H, p-value=0.03, Permanova), supporting that patients with high PSA or early metastatic PCa after prostatectomy display the strongest alterations in their gut microbiota".

β -diversity was also tested in the two mouse models and we found significant differences in both as described in the new results section.

Page 5 Lines 185-192: "As in the human prostate cancer fecal samples, in the TRAMP-2 model we also noted reduced alpha-diversity for the microbiota profiles collected during active growth and at end point compared to the baseline microbiota (Fig.2C, $p=3e^{-4}$, $p=2e^{-4}$, respectively, Welch's t test). Principal component analysis using the Bray-Curtis beta-diversity index revealed similarities between the fecal microbiota of animals at the stage of active growth and end point (Fig.2D). In contrast, the beta-diversity of both of active growth and end point microbiotas were also significantly distinct from those of animals at baseline or with minimal tumour mass (Fig. 2C, $p=0.001$, Permanova)".

Page 6 Lines 205-208: "We observed that the microbiota from animals with *Pten*^{-/-}; *Rb1*^{-/-} tumours was significantly different from the microbiota of no tumour controls using the Bray-Curtis beta-diversity principal component analysis (Fig.2G, p-value=0.02, Permanova), on par with the same analysis in the TRAMP-C2 model (Fig.2D)".

Reviewer #2

This is a very interesting study and a substantial piece of work. I have some questions and suggestions, which I feel could improve clarity and should be addressed before the manuscript may be accepted for publication.

1. The description of human samples/patients cohorts in the methods section lacks more details. Where/when were the patients recruited and samples collected (hospital, date/year)? Basic clinic information for all individual patients is missing, and should be consistently reported for both patient sets so that they can be more clearly compared (age, ethnicity, PSA level at diagnosis/treatment, TNM stage and Gleason Grade at diagnosis and at RP).

We thank reviewer #2 for the comments and questions that significantly contributed to strengthen the manuscript.

As suggested by the reviewer #2 we provided additional details describing the clinical and pathological characteristics of prostate cancer for all patients who provided fecal samples for the different experiments presented in the manuscript (supplementary tables 1 to 5). Patients were recruited between February 2015 and June 2017. This was added to the manuscript, Page 16, line 530-531.

2. Have any of the patients taken antibiotics – and when, as compared to time of fecal sampling? Most often antibiotics is used prior to prostate biopsy, and this could affect the gut microbiome?

All patients received our usual clinical protocol antibiotics at prostate biopsy and at radical prostatectomy. These precisions were added to the material and methods section.

Page 16, lines 523-529: “All patients in the pre-prostatectomy group (Fig 1A -D) received one dose of 500mg Ciprofloxacin on the day of the prostate biopsy which was performed on average 22.8 ± 1.43 weeks prior to sample collection in all pre-prostatectomy patients. While antibiotics may affect the microbiota, all patients had a similar exposure. As for patients with or without PSA failure after radical prostatectomy (Fig 1 E-H), all patients received one dose of intravenous Cefazolin (2g) at the beginning of surgery. All fecal samples from this cohort were collected several years after surgery”.

3. The quality of the 16sRNA seq data is difficult to assess. Not clear how many reads were generated for each sample (total/mapped), before QC and after QC (both human and mouse data).

We thank reviewer #2 for his insight regarding 16sRNA sequencing quality assessment. We added general information describing the total number of reads obtained from the sequencing run

and at the different steps of the bioinformatic analysis for both human and mouse fecal samples in the material and methods section.

Page 17, lines 573-577: “The methods used to analyse the bacterial taxonomic profiles of the murine gut microbiota are described in Anhê et al, 2017.⁴ All samples were rarified to the same sampling depth of 3 676 reads. Before quality check, a mean of 65 720 raw reads were generated for each sample, whereas after merging, quality filtering, and mapping steps, the average number of reads per sample was 14 854.”

4. I do not feel, that the results in figure 1 are convincing. There seems to not be a consistent pattern in the two patient sets for neither Firmicutes or Bacteroidetes abundance, nor for alpha diversity. Another major limitation in this part is that it is underpowered, with very small numbers in some of the groups, which is particularly problematic given the significant heterogeneity (wide error bars) and the inconsistency between the two patient sets.

The PCA plots in Suppl Fig 1 also seems to not shown any clearly distinct separation between the 3 subgroups defined for each patient set. The two patient sets also seem to be rather different, with samples collected prior to RP versus at BCR (where the whole prostate has been removed) – or at least that is how I understand it?

We thank reviewer #2 who raised several points regarding the data presented in the initial Fig.1. It made us realize that we did not provide a clear explanation for the two cohorts of patients, nor did we explain well the rationale for the two mouse models (see reply to comment 7 below) and their relation with the patients’ cohorts. The objective, here, was to examine the gut microbiota in relation with markers of cancer aggressiveness in two independent cohorts of patients who are, within each cohort, at a similar care pathway timepoint, thereby minimizing variation in potential microbiota confounders, such as antibiotics. We also agree with reviewer #2 that some groups had a small number of patients in the initially submitted Fig.1.

In the revised Fig.1 we present first (Fig. 1A-D) the results on fecal microbiota collected before prostatectomy (PR) according to the volume of the cancer measured on the subsequent prostatectomy specimen (5.0±0.5 weeks later): low (3.2 ± 1.4 g), medium (7.4 ± 1.4 g) or high (18.7 ± 12.7 g) (Fig.1A). **We were able to add samples from 20 new patients: 12 with high, 6 with medium and 2 with low tumor volume bringing the whole pre-PR cohort to 62 patients.** The 16S rRNA metataxonomic analysis showed no difference at the phylum level, but a significant reduction of the Shannon diversity index for patients in the highest tumor volume compared to patients in the low tumor volume group (Fig. 1C), in these patients with an otherwise similar localized PCa (Suppl Table 1).

We then present the second cohort (Fig. 1 E-H) with fecal samples collected years after prostatectomy (5.2 ± 1 years) in patients with undetectable PSA without biochemical recurrence (BCR), patients with very low PSA values (0.17 ± 0.02 ng/mL) and patients with significantly high

PSA values (2.96 ± 0.46 ng/mL) (Fig.1E). The latter group are patients with early metastasis as detected today with more sensitive TEP-PSMA nuclear imaging (explanations included in the new results section). **We were also able to add samples from 20 new patients to this cohort as well: 17 in the no BCR group, 2 in the low PSA and 1 in the High PSA Fig. 1 A-C (total 47 patients).** We observed significant differences at the phylum level in the gut microbiota of patients with BCR compared to no BCR. Noticeably, *Firmicutes* relative abundance was significantly higher in patients with high PSA compared to patients with low or undetectable PSA (Fig.1F, red bars). Of note a similar increase was also noted in the mouse model with loss of both PTEN and Rb (Fig. 2F), a model that replicate patients metastatic PCa where loss of Rb protein is common (PMID: 30553611 cited in the revised manuscript among many publications on the subject). The Shannon diversity index was also reduced in the metastatic (High PSA) group compared to the no BCR or low PSA BCR groups. Beta-diversity analysis also showed a significant shift in the high PSA population compared to low PSA or no BCR patients (Fig. 1H, p -value=0.03, Permanova), supporting that patients with high PSA or early metastatic PCa after prostatectomy display the strongest alterations in their gut microbiota.

Reviewer #2 also noted that the two patient sets (pre-RP versus post-RP BCR) seemed to have rather different microbiota profiles. Indeed, there were less *Firmicutes* and more *Bacteroidetes* in the patients pre-RP (Fig.1B) compared to patients with no BCR years post-RP (Fig.1F). It is not clear if the radical prostatectomy alone accounted for these changes in the microbiota since we lack the longitudinal microbiota analysis and the proper controls to establish this association. Nevertheless, the data show that there was no change at the phylum level with increasing PCa volume (g) but a significant reduction of alpha-diversity (Fig. 1C, $p=0.02$, Welch's t test) and no difference in beta-diversity between the three groups pre-RP. To the contrary, patients post-RP with metastasis (High PSA BCR) had a significant increase of *Firmicutes* (Fig.1F, red bars, $p=0.04$, Welch's t test) and reduced *Bacteroidetes* (Fig.1F, green bars, $p=1.5e^{-6}$, Welch's t test) compared to the no BCR patients. They also had a more pronounced reduction in alpha-diversity and a significant difference in beta-diversity compared to the no BCR, supporting our conclusion to this section:

Page 5, lines 169-172: "Beta-diversity analysis also showed a significant shift in the high PSA population compared to low PSA or no BCR patients (Fig. 1H, $p=0.03$, Permanova), supporting that patients with high PSA or early metastatic PCa after prostatectomy display the strongest alterations in their gut microbiota".

As also pointed out by reviewer #2, the microbiota of clinical samples displayed large inter-individual heterogeneity, especially for the pre-RP patients but to a lesser extent in the post-RP patients. Arguably, this variation between individuals could be impeding our effort to observe consistent changes in specific phylum. More in-depth analysis of the gut microbiota is needed to fully understand the specific changes in the gut microbes as a consequence of PCa development. For example, shotgun metagenomics would be useful to identify the specific bacterial genes consistently associated with PCa.

To address the issue with added the following in the Discussion section, page 12, lines 407-415:

“Although we observed a consistent loss of alpha diversity in patients and mouse models of PCa, we did not find a conserved set of taxa systematically altered in PCa samples, except for a significant increase in *Firmicutes* in fecal samples from early metastatic PCa (high PSA BCR) and in the *Pten*^{-/-}; *Rbl*^{-/-} mouse tumours. Patient-to-patient variability is a common feature of human gut microbiota studies and is not unique to our study^{5,6}. As previously reported⁷, the baseline microbiota was also different between mouse models, which also possibly impaired our ability to identify consistent changes in gut taxa as a result of PCa development. Repeated measures, paired analysis and whole-genome sequencing could help improve our capacity to identify consistent changes in gut taxa as a result of PCa development”.

5. Figure 1A and 1E are not very informative, would be useful with more clinical information about these patient sample sets. See my comment no 1.

As per the reviewer #2 suggestion, we added additional details in the description of human samples and included the clinical information for all the individual patients in supplementary tables 1 and 2.

6. There are samples from 8 mice in figure 2B/E, but 12 mice in figure 2A/2D. How come? Any selection?

As rightfully pointed out by reviewer #2, a total of 12 mice were used to measure TRAMP-C2 tumor development and specific gut microbes by qPCR for each PUFA treatment groups. However, to minimize costs, 8 representative fecal samples were sent for 16S rRNA sequencing experiments for each treatment group and timepoint. All microbiome data is presented.

7. The data in figure 2B (mouse data) seems to conflict with the observations in fig 1 (patient data). In patients, the authors report a trend toward higher abundance of Firmicutes and lower abundance of Bacteroidetes in patients with higher PSA/tumor burden, but the opposite pattern is seen in the mice (higher tumor burden associated with lower Firmicutes and higher Bacteroidetes). In fact, there seems to be a pattern through the whole manuscript, that only a selected part of the results presented are consistent with the overall conclusions made and final “biological model” presented.

In the new Fig. 2 we present, in the same order, analyses similar to those performed in the patients’ samples of Fig.1: tumour volume, microbiota phylum, alpha-diversity and beta-diversity to allow an easier comparison between patients’ cohorts and mouse experiments. We also clarified the objective of each set of mouse experiments in the results section.

In the TRAMP C2 model, we analysed the gut microbiota of the same mice at different time-points of tumour growth: minimal mass, active growth (average 400 mm³) and end-point (2,000 mm³). We observed a significant reduction of *Firmicutes* and increase of *Bacteroidetes* in active growth and end-point status.

The mouse models *Pten* knock-out and knock-out of both *Pten* and *Rb1* were used to compare tumours with increasing aggressiveness. They were tested compared to no tumour in 8 mice each at the 13 weeks' time-point when tumours with loss of both *Pten* and *Rb1* (a common trait of metastatic PCa) had an average volume of 400 mm³. There, we observed an increase of *Firmicutes* in the double knock-out of *Pten* and *Rb1* and a reduction of *Actinobacteria*.

However, in the two sets of experiments we observed a significant reduction of alpha-diversity and significant difference in beta-diversity in the larger tumours and in the more aggressive cancers leading us to conclude in the sub-section of PCa mouse models the following:

Page 6, lines 212-214: “These results support our observations in the two PCa patient cohorts that tumour volume and especially cancer aggressiveness are linked to the strongest fecal microbiota alterations (Fig.1C, G)”.

It is important to note that the TRAMP C2 experiments and the PTEN and Rb loss models were performed in two different laboratories. Several studies have described significant differences in the gut microbiota of genetically identical mouse models depending on the vendor⁷ (PMID 25675094) let alone the housing facility. Among all the different gut microbiota changes measured in the three different prostate cancer mouse models, the reduction in alpha-diversity was consistently correlating with tumor development and aggressiveness in our experiments. As stated before, the taxa contributing to this effect of reduced alpha diversity varied between the different mouse models. We have added sentences in the discussion section to clearly address the discrepancy between the different mouse models regarding changes in *Firmicutes* and *Bacteroidetes*. (see above in reply to comment 4).

Page 12, lines 407-415: “Although we observed a consistent loss of alpha diversity in patients and mouse models of PCa, we did not find a conserved set of taxa systematically altered in PCa samples, except for a significant increase in *Firmicutes* in fecal samples from metastatic PCa (High PSA BCR) and in the *Pten*^{-/-}; *Rb1*^{-/-} mouse tumours. Patient-to-patient variability is a common feature of human gut microbiota studies and is not unique to our study^{5,6}. As previously reported in the literature⁷, the baseline microbiota was also different between mouse models, which also possibly impaired our ability to identify consistent changes in gut taxa as a result of PCa development. Repeated measures, paired analysis and whole-genome sequencing could help improve our capacity to identify consistent changes in gut taxa as a result of PCa development”

8. How do the authors explain the opposite direction of changes in abundance of *Firmicutes* in the different mouse models (Fig 2B vs 2E), and the difference in the PCA plots in the different mouse models (compare Fig 2C and 2F).

As reviewer #2 pointed out, the changes in the relative abundance of *Firmicutes* and *Bacteroidetes* is not consistent across our mouse models and prostate cancer patients. The TRAMP C2 tumours are in fact the result of inactivation of the *Rb1* as stated in the discussion:

Page 12, lines 394-395: “SV40 large T antigen transgenic expression in TRAMP-C2 prostate tumour cells alters the *TP53* and *Rb1* signaling pathways leading to constitutive tumour cell growth⁸”.

Thus, the TRAMP C2 experiments are assessing the effect of tumour volume of a more aggressive tumour that may be similar to the tumours with knock-out of both *PTEN* and *RBI* and different from the PCa patients with localized cancer in the pre-RP cohorts. The reduction of *Firmicutes* is observed in the active growth and end-point, beyond the 400 mm³ size of the *PTEN* and *RBI* loss tumors at 13 weeks. Nevertheless, the PICRUST analyses of the large tumours in the three mouse models identified three common enzymes involved in the metabolism of long-chain Fatty Acids which was the basis for the second part of the study for which PCa growth and gut microbes were targeted by Omega-3 fatty acid supplementation (main Figure 4E, F and supp. Figure 2).

9. What is the Gleason score for each donor (Fig 3C). How were the donors selected? Were their microbiota composition known beforehand (16S rRNA seq data)?

Donors were selected to represent a variety of estimated tumor mass and other clinic-pathological measures of aggressiveness. The ISUP grade group for each donor used in Fig 3C is shown in the new supplementary table 3 and the corresponding Gleason score is described in the table’s legend. These patients had their microbiota profiled and most importantly provided enough fecal material for fecal transplantation into recipient mice and were thus suitable for the FMT experiments. Moreover, as per comments from another reviewer, we increased the number of FMTs from different patients in the new submission (see new Fig.3).

10. Add dots for individual mice in figure 3D, so easier for readers to assess variability. Error bars show SEM, which will visually minimize variation.

As per reviewer #2’s suggestion we added dots for figure 3D.

11. It would be interesting to compare the results in Figure 4A+B, by performing similar analyses using the data for the two patient data sets shown in figure 1/5. Is the *PTEN/RB1* status known for the patient tumors?

We thank reviewer #2 for this insightful comment. Unfortunately, we do not have the *PTEN/RB1* status on the primary PCa from the two patient data sets. That being said, many studies very well summarized in an extensive review of the subject in *Nat Rev Urol 2021* PMID: 33328650⁹ have

shown that PCa metastasis mostly originate from small clones within the primary tumour and may evolve under the pressure of the metastatic environment. Thus, the analysis of *PTEN* and *PTEN/RBI* protein expression on the primary tumor of patients with metastasis (i.e. High PSA BCR) would not be helpful. On the other hand, studies on metastasis have repeatedly shown that metastatic PCa have loss of *PTEN* and *RBI*⁹ PMID: 30553611.

12. The results shown in figures 5D-F are difficult to interpret. How can you tell the cause of down/upgrading from biopsy to RP? Systematic TRUS biopsy is notoriously error prone due to subsampling errors, leading to re-classification of say >40% of PC tumors at radical prostatectomy. Hence, it would be crucial to know if the clinical characteristics of the patients in the two groups (HOSO vs. MAG-EPA) are comparable – if not, it may not be surprising that they differ in the proportion of up/down-graded cases.

The “cause” of the down/upgrading between biopsy and RP is not easy to answer. However, with the findings of the butyrate as a potential mediator, several hypotheses can be considered that could result on the selective regression of the most aggressive pattern 4 cancer within a short period. We however agree with reviewer #2 that the Group Grade re-classification between the diagnostic biopsy and the prostatectomy is often viewed as a sampling error at the biopsy. However, compared to placebo, in the clinical trial of 130 patients we saw a significant reduction of the up-grading and increased down-grading in the MAG-EPA group (see below). Moreover, the up-grading observed in the placebo arm was similar to the rate observed in non-trial comparable patients from our institution and that of studies using the Grade Group found to result in less discordance between the biopsy and the prostatectomy grading. As for the 41 patients who provided fecal samples within the study before treatment and at prostatectomy, they were evenly distributed between HOSO and MAG-EPA and there was no difference between the two groups in terms of Group Grade and stage at the prostatectomy (Suppl Table 5). However, the significant reduction of up-grading and increase in down-grading was also observed in this sub-group of patients.

Below are extracts from the results and the discussion of the clinical study in 130 patients of the randomized trial currently under peer review.

“One of the most important determinants of prostate cancer aggressiveness and risk of progression is the cancer grade. However, prostate cancer grading is complex due to the frequent multifocality of tumors with distinct aggressive potential within an individual prostate (Cooper Nat Genetics 2015). Since 2014, the International Society of Urological Pathology (ISUP) new system of five prostate cancer Group-Grades (GG) is based on the proportion of the three Gleason patterns 3, 4 or 5¹⁰⁻¹². GG 1 tumors with only Gleason pattern while GG 2, 3 and 4 are defined by the proportion of pattern 4 (<50%, >50% and 100% respectively), and GG 5 have pattern 5. This new

classification showed less variation between the grade of the diagnostic biopsy and that of the prostatectomy specimen than the traditional Gleason score^{13,14}.”

In the phase IIB MAG-EPA vs Placebo pre-prostatectomy (under review in Nat Com), using the new ISUP GG classification, we observed significantly more down-grading and less up-grading of cancer at prostatectomy in the MAG-EPA group compared to placebo (see Table below). The subset of patients part of the microbiome study showed the same changes although in a smaller group of 21 patients in each arm (Suppl Table 5).

	Placebo N = 65 *	MAG-EPA N = 63 *	between groups
Grade reclassification, n (%)			P= 0.01 †
Upgrade	16 (24.62)	9 (14.29)	
Same	41 (63.08)	37 (58.73)	
Downgrade	8 (12.31)	17 (26.98)	

Previous studies showed that patients with up-graded disease exhibit more aggressive pathological features than concordant tumors and a higher risk of biochemical recurrence after radical prostatectomy¹⁵. When comparing the trial patients to 348 non-trial patients treated at our institution and meeting the trial entry criteria, we observed up-grading in 27% of patients, similar to the 25% of the placebo arm of the trial (data not shown) and to the 30% in the placebo arm of the present study. This up-grading was also similar to the 24% reported in 396 patients of a recent Australian study comparing ISUP GG to Gleason Score¹³. The 12.3% downgrading observed in the placebo arm of the trial was also similar to the 8% downgrading reported previously^{13,14}, suggesting that the control arm of the trial was representative of the average radical prostatectomy population.

With the exception of GG 5 who have Gleason pattern 5 in the tumor mix, GG 1, 2, 3 and 4 are defined by the proportion of Gleason pattern 4. A down-grading or lack of up-grading between the biopsy and the prostatectomy is due to a reduction of the percent of pattern 4, suggesting that high EPA prostate tissue level affect more selectively this cancer subtype within the prostate. Such an effect within a short time period and more specifically on Gleason pattern 4 cancer would support previous epidemiologic observations of an effect of LCn3 diet on more aggressive cancer.

Whether Gleason pattern 4 are distinct from pattern 3 within the same prostatectomy specimen was addressed recently by Sowalsky *et al.* who performed whole exome sequencing and transcriptomic profiling of laser-captured adjacent pattern 3 and 4 in patients with Gleason Score 7 (GG2/3) cancers. Large numbers of unique mutations in Gleason pattern 3 and pattern 4 tumors showed that Gleason pattern 4 were not derived from pattern 3, although no consistent genomic alterations could distinguish between them. However, pairwise transcriptome analyses identified

activated c-Myc pathway and decreased P53 activity in Gleason pattern 4 versus adjacent pattern 3, a hallmark of potentially more aggressive tumors¹⁶.

We have added the following text in the results section:

Page 9, lines 314-324: “PCa grade is one of the most important determinants of PCa aggressiveness and risk of progression. However, PCa grading is complex due to the frequent multifocality of tumors with distinct aggressive potential within an individual prostate¹⁷. The new International Society of Urological Pathology (ISUP) system of five prostate cancer Group-Grades (GG) is based on the proportion of the three Gleason patterns 3, 4 or 5¹⁰⁻¹². GG 1 tumors have Gleason pattern 3 only, while GG 2, 3 and 4 are defined by the proportion of pattern 4 (<50%, >50% and 100% respectively), and GG 5 have pattern 5. This new classification showed less variation between the grade of the diagnostic biopsy and that of the prostatectomy specimen than the traditional Gleason score^{13,14}. Previous studies showed that patients with up-graded disease exhibited more aggressive pathological features than concordant tumors and a higher risk of biochemical recurrence after radical prostatectomy¹⁵”.

Interestingly, we observed that grade group upgrading in the present study of 41 patients was associated with a concomitant increase in the blood PSA levels over the 7 weeks of the treatment before RP (Reviewer Figure 1), possibly supporting the biological relevance of grade group reclassification.

Reviewer Figure 1. Change in PSA in relation to PCa grade group reclassification in the 41 patients of the present study. Log₂-transformed change in blood PSA of patients corresponding to main Fig.5D was calculated in relation to their prostate tumour grade change (Gleason score from histopathological analysis of prostate tumour at surgery (Chx) compared to histological score of the biopsy of study enrolmen (BL). n=41, p-values from Kruskal-Wallis test are shown. We wish not to publish these results in the final manuscript since this observation need further confirmation in a larger cohort.

13. In continuation of this, do the authors suggest that treatment with MAG-EPA over a period of only 7 weeks can alter the gut microbiota (Fig 5F indicates that this is rather heterogeneous) and

in turn that this will alter the Gleason grade of the PCa tumor (Fig. 5E)? Or is it perhaps more likely, that the reclassification is mainly (or exclusively) due to the error-prone biopsy procedure (misclassification of Gleason score). Would be interesting to know what the change in the microbiome is over time in patients not receiving a supplement (negative controls)

The reclassification of Group Grade has been addressed in reply to comment 12 above. Our data supports that 7 weeks of MAG-EPA supplementation was sufficient to modulate the levels of *Ruminococcaceae* in PCa patients compared to the placebo HOSO.

As suggested by the reviewer, we measured the level of *Ruminococcaceae* in a cohort of 12 new PCa patients that had provided a fecal sample the day before their radical prostatectomy and another sample 2 months before on average and received no supplement. In contrast with the MAG-EPA group and similarly to the placebo group in Fig. 5F, we did not observe a significant difference in the level of *Ruminococcaceae* between the two fecal samples from patients not receiving any supplement. The result is shown below (Reviewer Figure 2). We wish not to have this data included in the manuscript since these patients did not participate into the clinical trial. Therefore, description of this additional dataset would increase the complexity of the manuscript perhaps futilely. However, we could include them if found important by the editors.

Reviewer Figure 2. Comparing *Ruminococcaceae* levels from fecal samples of untreated PCa patient or the placebo (HOSO) and MAG-EPA groups from the clinical trial. A) Relative abundance of *Ruminococcaceae* measured by 16SrRNA sequencing using fecal DNA sample collected from PCa patients 9 weeks prior (Baseline) and the day before their radical prostatectomy surgery (Pre-surgery). $p=0.33$, Paired-Wilcoxon test, $n=12$. **B)** Using DNA extracted from fecal samples corresponding to main Figure 5 E), we used 16S rRNA metataxonomic to compare the profiles corresponding to men before and after 7.2 ± 0.37 weeks of MAG-EPA ($n=21$) supplementation or placebo ($n=20$). The relative abundance of sequences corresponding to *Ruminococcaceae* is shown, $p \le 0.05$, Paired-Wilcoxon test. Graphs are mean \pm SEM.

14. Furthermore, it would be interesting to know if the relative abundance of Ruminococcaceae in samples at baseline and at RP, respectively, correlate with the true Gleason Grading for that patient (i.e. determined at RP, not at TRUSbx).

There was no association between the relative abundance of *Ruminococcaceae* with grade (ISUP or Gleason score) at RP or at baseline.

15. Matched data is available for the results presented in figure 6A-F. Would be helpful to also show those rather than just the averages, to better understand the possible heterogeneity in the effects. What is the rationale for not including a no-treatment group as a control – could reveal the level of variation/heterogeneity in signals?

In the Figure 6A-F, we performed comparison of HOSO and MAG-EPA-related SCFA levels using the paired data and the average levels after treatment. We found a trend for isovaleric acid ($p=0.06$) and a significant difference in butyric acid levels, respectively. As mentioned before, the clinical trial (NCT02333435) was not designed with a no-treatment group but rather a placebo group for better internal control. However, we found similar data for levels of *Ruminococcaceae* in a small cohort of 12 patients compared to the levels of *Ruminococcaceae* observed in the placebo group, the data is presented above as a figure for reviewers only (Reviewer Figure 2).

16. I do not feel the compared groups in figure 6G are meaningful, for the same reasons as stated in my comments 12-13.

We understand that grade group reclassification is generally thought to be a consequence of the partial sampling occurring at the biopsy procedure. However, our data from a randomized controlled trial showed that a dietary supplementation significantly impacted the grade group reclassification in prostate cancer patients. Therefore, our observations suggest that all grade group reclassification might not be explained by a bias of the biopsy procedure. We are hopeful that the explanations provided in the reply to comment 12 will answer these concerns.

Reviewer #3 (Remarks to the Author):

General Comments:

1. The study of gut microbiota is essential for gaining a more thorough understanding of the interactions of the microbiota with oncogenesis, tumor progression and cancer therapy. Therefore, this study provides new knowledge on the potential role of the status of retinoblastoma (Rb) gene expression and the gut microbiota on prostate cancer.
2. Figures 4 and 5 are perhaps the most interesting in the entire study. The use of the three mouse models, the in-depth analysis of differences between their microbiota, the implication of loss of Rb expression and/or function in altering microbiota and how this could modulate tumor growth and the response of tumor growth to treatment with different long-chain polyunsaturated fatty acids would make a very interesting and insightful manuscript on its own.
3. The inclusion of Figure 1 in this study is unclear. While an in-depth analysis of these human fecal samples has great potential, this was not realized in the overall experimental design of the manuscript. Furthermore, there is a concern that the formula used to determine low, medium and high tumor burden might have skewed the data in this study (as discussed below). In addition, the link between the data in Figure 1 and the remainder of the paper is tenuous. Even the bacteria analyzed (Firmicutes and Bacteroidetes) appear to be of minor importance in the mouse models where Parabacteroidetes, Lactobacillus, and Ruminococcaceae take center stage. This is confirmed in the experimental design where (a) Rb status in mouse tumors was known; Rb status in human tumors was not known, (b) mouse fecal samples were thoroughly analyzed using multiple experimental and analytical approaches (figures 3, 4, SF2, SF3 part1, SF3 part2, SF4, and 5); human fecal samples were not, etc. The only exception appears to be the clinical study where MAG-EPA PUFA or HOSO was administered to patients prior to prostatectomy (Figures 5D-5F, S5) and human fecal material before and after prostatectomy were analyzed. Therefore, it might be advantageous to consider the value of the human fecal samples collected in Figure 1 and design a comprehensive study that would optimize their value.

We thank reviewer #3 for these very insightful comments that made us realize that we did not provide a clear explanation for the two cohorts of patients, nor did we explain well the rationale for the two mouse models (see reply to comment 7 below) and their relation with the patients' cohorts.

We removed the original Figure 1 panels B-C and F-G and placed the original supplementary Figure 1A and C in main Figure 1, to improve interpretation of the main finding regarding loss of alpha-diversity.

Patients' cohorts: In the revised Fig.1 we present first (Fig. 1A-D) the results on fecal microbiota collected before prostatectomy (RP) according to the volume of the cancer measured on the subsequent prostatectomy specimen (5.0±0.5 weeks later): low (3.2 ± 1.4 g), medium (7.4 ± 1.4 g) or high (18.7 ± 12.7 g) (% of cancer area multiplied by weight (g) of the radical prostatectomy specimen) (Fig.1A). **We were able to add samples from 20 new patients: 12 with high, 6 with medium and 2 with low tumor volume bringing the whole pre-PR cohort to 62 patients.** The 16S rRNA metataxonomic analysis showed no difference at the phylum level, but a reduction of the Shannon diversity index for patients in the highest tumor volume compared to patients in the low tumor volume group (Fig. 1C), patients with otherwise localized PCa (Suppl Table 1). Beta-diversity was also analysed following the request of another reviewer and showed no change between the 3 groups.

We then present the second cohort Fig. 1 E-H with fecal samples collected years after prostatectomy (5.2±1 years) in patients with undetectable PSA without biochemical recurrence (BCR), patients with very low PSA values (0.17±0.02 ng/mL) and patients with significantly high PSA values (2.96±0.46 ng/mL) (Fig.1E). The latter group are patients with early metastasis as detected today with more sensitive TEP-PSMA nuclear imaging (explanations included in the new results section). **We were also able to add samples from 20 new patients to this cohort as well: 17 in the no BCR group, 2 in the low PSA and 1 in the High PSA Fig. 1 A-C (total 47 patients).** We observed significant differences at the phylum level in the gut microbiota of patients with BCR compared to no BCR. Noticeably, *Firmicutes* relative abundance was significantly higher in patients with high PSA compared to patients with low or undetectable PSA (Fig.1F, red bars). Of note a similar increase was also noted in the mouse model with loss of both PTEN and Rb (Fig. 2F), a model that replicate patients metastatic PCa where loss of Rb protein is common (PMID: 30553611 cited in the new manuscript among many publications on the subject). The Shannon diversity index was also reduced in the early metastatic (High PSA) group compared to the no BCR or low PSA BCR groups. Beta-diversity analysis also showed a significant shift in the high PSA population compared to low PSA or no BCR patients (Fig. 1H, p-value=0.03, Permanova), supporting that patients with high PSA or early metastatic PCa after prostatectomy display the strongest alterations in their gut microbiota.

In view of the significant reduction in fecal butyrate by 7 weeks of MAG-EPA treatment associated with a reduction of the prostate cancer grade at prostatectomy, we tested fecal butyrate levels in the cohort of post-prostatectomy patients (Fig.1E). This **new data** is detailed in new Fig.6 H. We found an increase of the butyrate levels in patients with early metastatic cancer (High PSA BCR) compared to the no BCR or low PSA group. This validation is an independent patient cohort at a later disease stage supports the hypothesis that altered microbiota metabolism drives PCa aggressiveness.

Mouse experiments: In the new Fig. 2 we present, in the same order, analyses similar to those performed in the patients' samples of Fig.1: tumour volume, microbiota phylum, alpha-diversity and beta-diversity to allow an easier comparison between patients' cohorts and mouse experiments. We also better defined the objective of each set of mouse experiments in the results section.

In the TRAMP C2 model, we analysed the gut microbiota of the same mice at different time-points of tumour growth: minimal mass, active growth (average 400 mm³) and end-point (2,000 mm³). We observed a significant reduction of *Firmicutes* and increase of *Bacteroidetes* in active growth and end-point status.

The mouse models *Pten* knock-out and knock-out of both *Pten* and *Rb1* were used to compare tumours with increasing aggressiveness. They were tested compared to no tumour in 8 mice each at the 13 weeks' time-point when tumours with loss of both *Pten* and *Rb1* (a common trait of metastatic PCa) had an average volume of 400 mm³. There, we observed an increase of *Firmicutes* in the double knock-out of *Pten* and *Rb1* and a reduction of *Actinobacteria*.

However, in the two sets of experiments we observed a significant reduction of alpha-diversity and significant difference in beta-diversity in the larger tumours and in the more aggressive cancers leading us to conclude in the sub-section of PCa mouse models the following:

Page 6, lines 212-214: "These results support our observations in the two PCa patient cohorts that tumour volume and especially cancer aggressiveness are linked to the strongest fecal microbiota alterations (Fig.1C, G)."

It is important to note that the TRAMP C2 experiments and the PTEN and Rb loss models were performed in two different laboratories. Several studies have described significant differences in the gut microbiota of genetically identical mouse models depending on the vendor⁷. (PMID 25675094) let alone the housing facility. Among all the different gut microbiota changes measured in the three different prostate cancer mouse models, the reduction in alpha-diversity was consistently correlating with tumor development and aggressiveness in our experiments. As stated before, the taxa contributing to this effect of reduced alpha diversity varied between the different mouse models. We have added sentences in the discussion section to clearly address the discrepancy between the different mouse models regarding changes in *Firmicutes* and *Bacteroidetes*. (see above in reply to comment 4).

We have replaced "tumour burden" by "tumour volume and aggressiveness" in the title of the first section of the results and modified the sub-section "PCa patients" to incorporate the comments above (page 4, lines 126 and 128).

We also agree with the reviewer about the lack of consistent pattern of changes for *Firmicutes* or *Bacteroidetes* abundance when comparing prostate cancer patients and prostate cancer mouse models. On the other hand, our data in figure 1 and 2 support that prostate tumor development and aggressiveness is affecting the overall alpha diversity of the host's gut microbiota.

To address the issue with added the following in the Discussion section, page 12, lines 407-415: "Although we observed a consistent loss of alpha diversity in patients and mouse models of PCa, we did not find a conserved set of taxa systematically altered in PCa samples, except for a significant increase in *Firmicutes* in fecal samples from metastatic PCa (High PSA BCR) and in the *Pten*^{-/-}; *Rbl*^{-/-} mouse tumours. Patient-to-patient variability is a common feature of human gut microbiota studies and is not unique to our study^{5,6}. As previously reported⁷, the baseline microbiota was also different between mouse models, which also possibly impaired our ability to identify consistent changes in gut taxa as a result of PCa development. Repeated measures, paired analysis and whole-genome sequencing could help improve our capacity to identify consistent changes in gut taxa as a result of PCa development".

4. Overall, the Results Section tends to be descriptive. For example, in Figure 2, descriptions include, "showed a modulation", "were distinct", and "had similar fecal microbiota". In Figure 5, examples include "... there was an up-grading toward a more malignant cancer....., a down-grading toward a less malignant cancer...." and "significantly reduction". The term grading was frequently used, but grading scores were never provided, making it difficult to interpret the data. This study would be considerably strengthened by the use of analytical scientific language, by providing a rationale where appropriate and by succinct and quantitative analyses (in addition to p values) that support the interpretation of the results.

We thank the reviewer #3 for these comments and made changes accordingly to provide the quantitative values wherever possible in the results section.

A paragraph was added to explain the Group Grading system for prostate cancer in the results section just before the presentation of the results on the 41 patients who received MAG-EPA or a placebo before prostatectomy. The Grade Group reclassification: up-grading and down-grading for these patients is presented in Supplementary Table 5 in addition to the Fig.5.

Page 9, lines 309-324: "The clinical and biopsy characteristics at baseline and the pathological characteristics at radical prostatectomy of the sub-cohort of 41 PCa patients who agreed to provide fecal samples for research are shown in Supplementary Table 4 and 5, respectively. PCa grade is one of the most important determinants of PCa aggressiveness and risk of progression. However, PCa grading is complex due to the frequent multifocality of tumors with distinct aggressive potential within an individual prostate¹⁷. The new International Society of Urological Pathology (ISUP) system of five prostate cancer Group-Grades (GG) is based on the proportion of the three

Gleason patterns 3, 4 or 5¹⁰⁻¹². GG 1 tumors have Gleason pattern 3 only, while GG 2, 3 and 4 are defined by the proportion of pattern 4 (<50%, >50% and 100% respectively), and GG 5 have pattern 5. This new classification showed less variation between the grade of the diagnostic biopsy and that of the prostatectomy specimen than the traditional Gleason score^{13,14}. Previous studies showed that patients with up-graded disease exhibited more aggressive pathological features than concordant tumors and a higher risk of biochemical recurrence after radical prostatectomy¹⁵”.

Page 3, lines 117-119, the sentence: “Analysis of metabolites produced by the *Ruminococcaceae* family of bacteria in patients’ fecal samples treated with an omega-3 supplement identified a selective and significant reduction of butyrate” was changed to: “Analysis of metabolites produced by the *Ruminococcaceae* family of bacteria in patients’ fecal samples treated with an omega-3 supplement identified a selective reduction of butyrate”.

Page 4, lines 141-142, the sentence: “Interestingly, we observed significant reduction in Shannon diversity index for patients in the high tumor volume tertile was reduced compared to patient in the low tumor volume group (Fig. 1C).” was changed to: “The Shannon diversity index was reduced for patients in the highest tumor volume compared to patients in the low tumor volume group (Fig. 1C, $p=0.02$, Welch’s t test)”.

Page 5, lines 165-167, the sentence: “We also noted a significant reduction of *Bacteroidetes* relative abundance in BCR high PSA was significantly lower compared to no BCR patients (Fig.1F, green bars).” Was changed to: “We also noted that *Bacteroidetes* relative abundance in BCR high PSA was significantly lower compared to no BCR patients (Fig.1F, green bars, $p=1.5e^{-6}$, Welch’s t test)”.

Page 5, lines 167-169, the sentence: “Noticeably, as for patients with BCR after surgery, the gut microbiota of patients with high tumour burden displayed a significant reduction in the Shannon diversity index when compared to patients with low tumour burden (Fig.1H)”. Was changed to: “The Shannon alpha diversity index was also reduced in fecal samples from patients with high PSA compared to patients with undetectable or very low PSA levels (Fig.1G, $p=0.02$, Welch’s t test)”.

Page 5, lines 183-185, the sentence: “16S rRNA metataxonomic analysis showed a modulation of the phylum *Bacteroidetes* in fecal samples from exponentially growing and end point tumours compared to no tumour time point samples (Fig.2B)” was changed to: “When comparing fecal samples from actively growing and end point tumours to no tumour time point samples, 16S rRNA metataxonomic analysis revealed an increase in the *Bacteroidetes* phylum (Fig.2B, $p=8e^{-4}$, Welch’s t test)”.

Page 6, lines 188-192, the sentence: “Principal component analysis of Bray-Curtis beta-diversity index revealed that animals at the stage of active grow and end point had similar fecal microbiota (Fig.2C).” was changed to: “Principal component analysis using the Bray-Curtis beta-diversity index revealed similarities between the fecal microbiota of animals at the stage of active growth and end point (Fig.2D). In contrast, the beta-diversity of both of active growth and end point

microbiotas were also significantly distinct from those of animals at baseline or with minimal tumour mass (Fig. 2D, $p=0.001$, Permanova)”.

Page 5, lines 185-188, the sentence: “As in the human prostate cancer fecal samples, in the TRAMP-2 model, we also noted a significant reduction in the alpha-diversity for the microbiota profiles collected during active grow and at end point compared to the baseline microbiota (Fig.2D).” was changed to: “As in the human prostate cancer fecal samples, in the TRAMP-2 model we also noted a reduced alpha-diversity for the microbiota profiles collected during active growth and at end point compared to the baseline microbiota (Fig.2C, $p=3e^{-4}$, $p=2e^{-4}$, respectively, Welch’s t test)”.

Page 6, lines 205-208, the sentence: “The Bray-Curtis beta-diversity principal component analysis showed that the gut microbiota from animals with *Pten*^{-/-}; *Rb1*^{-/-} tumours was significantly different from the microbiota of no tumour controls (Fig.2G, p -value=0.02, Permanova), on par with the same analysis in the TRAMP-C2 model (Fig.2C).” was changed to: “We observed that the microbiota from animals with *Pten*^{-/-}; *Rb1*^{-/-} tumours was significantly different from the microbiota of no tumour controls using the Bray-Curtis beta-diversity principal component analysis (Fig.2G, p -value=0.02, Permanova), on par with the same analysis in the TRAMP-C2 model (Fig.2D).”

Page 9, lines 296-299, the sentence: “Interestingly, there was a significant reduction ($p=0.008$) in the level of sequences related to the *Ruminococcaceae* family (Fig.5B,C) in the two omega-3 PUFAs MAG-DHA- and MAG-EPA-treated animals compared to HOSO control.” was changed to: “Interestingly, the abundance of sequences related to the *Ruminococcaceae* family (Fig.5B,C) in the two omega-3 PUFAs MAG-DHA- and MAG-EPA-treated animals were significantly reduced ($p=0.008$, Welch’s t test) compared to HOSO control”.

Page 10, lines 326-328, the sentence: “When comparing the cancer grade on the diagnostic biopsy to the grade on the prostatectomy specimen, there was an up-grading toward a more malignant cancer in 8 patients, a down-grading toward a less malignant cancer in 5 patients and no change in the other 28 patients.” was changed to: “When comparing the cancer grade on the diagnostic biopsy with the grade on the prostatectomy specimen in the 41 patients, we observed an up-grading in 8 patients, a down-grading in 5 patients and no change in the other 28 patients”.

Page 10, lines 334-337, the sentence: “Noticeably, this lower frequency of PCa upgrade for the MAG-EPA group between pre- and post-treatment was associated with a significant reduction in the levels of sequences corresponding to the *Ruminococcaceae* ($p=0.046$) (Fig.5F).” was changed to: “Noticeably, compared to HOSO, MAG-EPA significantly reduced the abundance of sequences corresponding to the *Ruminococcaceae* ($p=0.046$, paired-Wilcoxon test) (Fig.5F) between pre- and post-treatment microbiota”.

Page 10, lines 343-345, the sentence: “Compared to placebo, PCa patients having received MAG-EPA supplement displayed a significant reduction of butyric acid ($p=0.04$) but not for the other SCFAs (Fig.6A-F), except for a similar trend of lower valeric acid levels (Fig.6D).” was changed to: “Compared to placebo, MAG-EPA reduced fecal butyric acid levels by a factor of

1.5 on average ($p=0.04$, Welch's t test). There was no effect on other SCFAs (Fig.6A-F), except for a similar reduction trend of valeric acid levels (Fig.6D)".

Page 10, lines 347-349, the sentence: "Remarkably, patients with PCa downgrading at surgery showed significantly less enrichment of fecal butyric acid levels than patients with no change ($p=0.002$) (Fig.6G)." was changed to: "Patients with PCa downgrading at surgery displayed less enrichment on average by a factor of 2 of fecal butyric acid levels than patients with no change ($p=0.002$, Welch's t test) (Fig.6G)".

Page 11, lines 372-374, the sentence: "We also observed a significant reduction in gut microbiota diversity in the two distinct mouse models that grew the largest tumours." Was changed to: "We also observed reduced gut microbiota diversity in the two distinct mouse models that grew the largest tumours, in particular those with loss of both PTEN and Rb, an alteration frequently observed in metastatic PCa patients^{18,19}".

Page 14, lines 450-452, the sentence: "Treatment of patients for 7 weeks on average before prostatectomy with an LCn3 MAG-EPA supplement compared to placebo was associated with a significant reduction of cancer upgrading and of fecal *Ruminococcaceae* levels." was changed to: "Treatment of patients for 7 weeks on average before prostatectomy with a LCn3 MAG-EPA supplement compared to placebo was associated with a reduction-of cancer upgrading and of fecal *Ruminococcaceae* levels".

Page 14, lines 468-470, the sentence: "Here, we observed a selective and significant reduction of fecal butyrate levels in the MAG-EPA-treated PCa patients in association of a significantly less cancer upgrading to a more malignant phenotype at prostatectomy." was changed to: "Here, we observed reduced fecal butyrate levels in the MAG-EPA-treated PCa patients in association with significantly less cancer upgrading to a more malignant phenotype at prostatectomy".

Page 15, lines 486-488, the sentence: "Our results showing a significant reduction of fecal butyrate levels after a short 7 weeks treatment with MAG-EPA LCn3 supplement may represent a promising and simple opportunity to improve the clinical efficacy of anti-CTLA-4 treatments." Was changed to: "Our results showing reduced fecal butyrate levels after a short 7 weeks treatment with MAG-EPA supplement may represent a promising and simple opportunity to improve the clinical efficacy of anti-CTLA-4 treatments".

Other Comments:

1. Materials and Methods, Human samples. Please provide additional details for the different patient cohorts/groups investigated in this study, for example:

Gleason Grade Group ≥ 2 localized PCa cohort:

As per the reviewer #3 suggestion, supplementary Tables 1 to 5 have been added to provide detailed clinical and pathological characteristics for all patients who provided fecal samples for the different experiments including the PCa grade and stage where appropriate.

a. Please define Gleason Grade Group ≥ 2 localized PCa as researchers outside the field may not understand the prostate cancer classification scale.

We added a definition of Gleason Grade Group ≥ 2 localized PCa in the material and methods. Page 15, lines 510-516: “Briefly, the International Society of Urological Pathology (ISUP) agreed on a system of five prostate cancer Grade-Groups based on the proportion of the three Gleason patterns 3, 4 or 5¹⁰⁻¹², which are associated with prostate cancer progression risk. Grade group 1 tumors with only Gleason pattern 3 are indolent and most patients are recommended active surveillance. Grade Groups 2, 3 and 4 are defined by the proportion of pattern 4 (<50%, >50% and 100% respectively), and Grade Group 5 have pattern 5, which is the most aggressive cancer”.

b. Please provide the number of patients providing fecal samples and range of weeks prior to prostatectomy.

Page 15, line 508-510. “A first panel of 62 prostate cancer (PCa) patients with Gleason Grade Group ≥ 2 localized PCa agreed to provide fecal sample 5.0 ± 0.5 weeks (mean \pm SEM) on average before they were treated by radical prostatectomy (RP) and on the day before the surgery”.

c. Provide the number of patients providing fecal samples and range of weeks after prostatectomy.

Page 15, line 516-518. “We also collected fecal samples from 47 PCa patients that had their prostate surgically removed by RP on average 5.2 ± 1.0 years (mean \pm SEM) before sample collection”.

d. Further subdividing these samples into additional categories would help with the interpretation of the data in Figure 1. Examples include:

These analyses were performed and, as expected because of small subcategory sample sizes, we did not find significant differences other than the tumor volume or BCR and therefore these results were not included in the initial submission in order to respect the word count and display items guidelines. However, here are some additional details.

- Gleason Score.

We did not find significant differences in relation to the Gleason score and these results were not included in the initial submission in order to respect the word count and display items guidelines.

- defining the range of PSA levels for each category which are currently only described as “low, medium, and high”

The range of PSA levels are described on page 16, lines 526-528. “These patients experienced different levels of cancer recurrence based on Prostate-Specific Antigen (PSA) serum levels; low PSA values (between 0.05-0.49 ng/mL) and 18 patients had clinically significantly high PSA values (above 0.50 ng/mL).”

- the range of % tumor verses benign areas in the tissue samples. The range of % tumor compared to
- Tumor volume – approximation as determined by the area under the tumor margins inked in by the pathologist
- other variables considered important to the interpretation of the data, e.g., age (as prostate volumes tend to increase with age and BPH content).

In the revised Fig.1 we present first (Fig. 1A-D) the results on fecal microbiota collected before prostatectomy (RP) according to the volume of the cancer measured on the subsequent prostatectomy specimen (5.0±0.5 weeks later): low (3.2 ± 1.4 g), medium (7.4 ± 1.4 g) or high (18.7 ± 12.7 g) (% of cancer area multiplied by weight (g) of the radical prostatectomy specimen) (Fig.1A). As for the grade and stage are presented in Suppl Table 1 for this cohort and in Suppl Tables 2-5 for the others. The age, body mass index and PSA values are also presented in these Tables.

e. Were the patients who provided fecal samples before radical prostatectomy the same group of men who also provided fecal samples after prostatectomy? If yes, could the “before” and “after” fecal samples from the same patient be compared? This linear analysis would increase statistical power considerably.

No, patients who provided fecal samples before and after radical prostatectomy are not the same individuals.

f. Was the range of fecal collection time prior to prostatectomy the same for the two groups described above as that for men who participated in the phase IIb/MAG-EPA trial, i.e., 5.2 weeks (± 4 days)?

The range was 5.0 ± 0.5 weeks vs 7.2 ± 0.37 weeks (mean \pm SEM).

Phase IIb double-blind randomized controlled trial group:

In a similar manner, please define the samples and other pertinent information for the patients who provided fecal samples for this current study.

Subgroup of 41 patients

In a similar manner, please define the samples and other pertinent information for the subgroup of 41 patients who consented to provide clinical data and fecal samples for microbiota high-throughput analysis:

- Including the additional 24 patients with biochemical recurrence.
- including the 3 patients with undetectable PSA levels

Using a table might be helpful in describing the different groups.

As suggested by the reviewer, we added supplementary tables 1 to 5 to better describe the cohorts.

2. Materials and Methods. Please provide a section that briefly, but succinctly, describes how the samples for Figures 1E through 1H were analyzed. This study uses the term “tumor burden” to describe the % cancer tissue / total prostate weight at surgery (g). “Tumor burden” is a clinical term that is more frequently used to describe the degree of metastasis in the body. Therefore, please provide a clear description of how the term is used in this study to prevent confusion. Alternatively, a term that pertains to the samples being analyzed, perhaps tumor tissue (volume) or tumor tissue (%) could be used.

We thank the reviewer #3 for these comments and made changes accordingly. The term “Tumor burden” was changed to “tumor volume and aggressiveness” in the revised Figures and in the manuscript.

3. Materials and Methods, Murine models and treatments. Regarding the experimental design:
a. For the TRAMP-2C study, please indicate that the cells were injected s.c. Similarly, please clarify how Pten^{-/-} ; Rb1^{+/+} and Pten^{-/-} ; Rb1^{-/-} cells were administered. Currently the text states that these cells were “injected in the flank”. This description could mean s.c., or it could also mean intramuscular.

We added to the text that all murine prostate tumor cells, including TRAMP-C2, Pten^{-/-} ; Rb1^{+/+} and Pten^{-/-} ; Rb1^{-/-} cells were injected subcutaneously in the flank of mice. Page 5, line 180 and 196.

b. Regarding the protocol for generating the tumors, please clarify whether (a) tumors were established first and then mice were treated or (b) tumor cells were injected after (or along) with treatment at the same time. The first approach would determine effect of treatment on established tumors. The second approach would determine the effect of treatment on the success of establishing tumors in addition to whether treatment would promote tumor growth (or regression).

Fatty acid gavage was started 2 weeks before tumor implantation and continued during tumor growth. We added this information in the revised material and methods section on page 18, lines 597-602: “To evaluate the effect of specific fatty acids on TRAMP-C2 tumour growth, mice were daily supplemented by oral gavage with 618 mg/kg of body weight of purified MAG-EPA (SCF Pharma, Ste-Luce, Canada), MAG-DHA (SCF Pharma), MAG-AA (Nu-Check-Prep Inc, Elysian, MN) or HOSO (SCF Pharma) before tumour cell injection and until animal sacrifice. The dose supplemented mimicked the clinical dose used in the clinical trial (3g/day of MAG-EPA, NCT02333435).”.

c. Mice were fed with a low-fat diet. Did this diet contain any of the specific fatty acids of interest?

We added the low-fat diet reference (Research Diet inc, #D12450H) in the material and methods (Page 18, lines 596) and a sentence describing the content of the specific fatty acids of interest: “0.2% of the total fat in the low-fat diet was arachidonic acid (AA) but the diet contained no DHA or EPA. (Page 18, lines 604-605).”

d. Mice were given 618 mg/kg of body weight of purified MAG-EPA daily via oral gavage. Does the mouse dose mimic the dose (3g/day of MAG-EPA) given to men participating in the Phase IIb clinical trial?

Yes, this information was added to the revised Materials and Methods. Page 18, lines 600-602: “The dose supplemented mimicked the clinical dose used in the clinical trial (3g/day of MAG-EPA, NCT02333435).”

e. For the FMT study, please indicate whether oral or intragastric gavage was used. If oral gavage was used, would this approach cause any alterations in bacterial survival rates and/or gene expression profiles as the donor microbiota pass through the environment of the stomach?

Intragastric gavage was used for the FMTs. This information was added to the revised Materials and Methods. Page 19, lines 612-613: “Receiver mice were given the PBS-resuspended feces from six independent PCa patients by intragastric gavage.”

f. Please provide a brief description of how the FMT material was prepared for gavage, what steps were taken to ensure a good representation of both aerobic and anaerobic bacteria in the specimens, and the administration regime used to maintain the human donor microbiota in recipient mice.

This information was added to the revised Materials and Methods. Page 19, lines 613-616: “Prior to the intragastric gavage into recipient mice, human fecal samples were culture tested for viability and FMTs were administered to the animals twice for two successive weeks and every time a drop of fecal suspension was added to the animal fur.”

g. Was the overall bacterial load in every human donor microbiota similar? If some samples had a low microbiota load while others had a high microbiota load, would this impact the results?

This is a very interesting point raised by the reviewer. Unfortunately, we did not directly measure the bacterial load of the donor’s fecal samples. However, as explained in f., the bacterial cultures derived from the donor’s fecal samples were tested for viable bacteria content using GIFU broth cultures and all donor samples displayed similar growth. Despite this, we observed significantly less culturable bacteria in receiver mice for FMT of high tumour volume donors (Fig 3E). We added a section of text that reflects our understanding of the data, whereby fecal samples from patients with high tumour volume have noticeably low alpha-diversity (Fig 1C), that could potentially lead to poor fecal microbiota transfer.

Page 7, lines 229-235: “However, there was no trend for increased TRAMP-C2 volume with human FMTs from patients with relation to increasing tumour volume (Fig.3D). On the other hand, we observed significantly less viable bacteria in fecal cultures of mice having received FMTs from patients with high tumour volume and low microbiota diversity (Fig.1C) compared to medium tumour volume FMTs (Fig.3E, $p=0.03$, Welch’s t test). This suggests that the low diversity of the microbiota could potentially impair fecal transplantation and thus explain the lack of increased TRAMP-C2 tumour volume for FMTs derived from high tumour volume patients”.

h. Please indicate the sex of the recipient mice. Males would provide the optimal supportive testosterone microenvironment for prostate tumor growth.

Male C57Bl6 mice were used, this information was added to the revised Materials and Methods. Page 18, line 590.

4. Results section, Figure 1A-1D. These panels compare differences in PSA and bacterial content of samples obtained from patients showing biochemical resistance (BCR). This analysis relies primarily on PSA levels. While increasing PSA levels remain useful as a biomarker for BCR, higher Gleason scores with lower PSA levels have also been observed in patients who are newly diagnosed with aggressive prostate cancer. Did the study criteria include MRI scans? If yes, then the scans would confirm biochemical recurrence as determined by increased lesion size and/or metastatic spread (i.e., increased tumor burden).

We thank the reviewer #3 for these comments and we specified in the manuscript that for revised Figure 1E-H, all patients had undergone previous radical prostatectomy several years before. After radical prostatectomy, increasing PSA levels is considered a reliable biomarker of cancer

recurrence, since the prostate cannot be not contributing to the raising PSA. Moreover, patients with an average PSA of 2.9 after prostatectomy have almost universally metastasis identified with the new TEP PSMA nuclear medicine imaging as added in the Results section.

5. Results, Figure 1E-1H. Figure 1E investigates tumor burden as defined by the % of cancer tissue / total prostate mass (g) at surgery. This analysis assumes that any changes in the weight of the total prostate correlates directly to the weight of a tumor in the prostate. However, the total weight of each patient prostate will be different; and total prostate weight will likely increase with age and contain differing amounts of BPH. Therefore, including the variables of weight, age and BPH content into the calculation could potentially “mask” the levels of cancer tissue in the whole prostate. Comparing % cancer area (and cancer stage?) verses benign area per biopsy (where n = the number of tissue slides analyzed) to total tumor volume as defined by the total area under the positive tumor margins inked on the biopsy tissue slices/pieces by the pathologist might be more informative and statistically significant. Similar comments pertain to the X axis in the analysis of bacterial content in 1F through 1H.

We thank the reviewer #3 for these comments. We clarified the calculation associated with the tumor volume data in the revised material and methods on page 16, lines 532-534. It is important to note that the tumor volume is calculated on the prostatectomy specimen performed on average 5.0 ± 0.5 weeks later and not on biopsies, as this standard clinical measure at prostatectomy is more reliable than any measure at biopsy. The detailed pathological information about the prostatectomy specimen is in the suppl Table 1. Consequently the % cancer X by the weight of the whole prostate in g gives a PCa volume in grams as introduced in the new results section.

Also, as correctly pointed out, other measures of cancer aggressiveness do exist at pathology. They are reported in the suppl Table 1. This information can now be used to help interpret the study findings.

We also clarified the calculation associated with the tumor mass data in the revised result section on page 4, lines 134-136: “Based on the surgical pathology report, fecal samples were separated in tertiles of low (3.2 ± 1.4 g), medium (7.4 ± 1.4 g) or high (18.7 ± 12.7 g) PCa tumour volume (% of cancer area multiplied by weight (g) of the radical prostatectomy specimen)”.

6. Results, Figure 2. Fig. 2A lacks a quantitative description of the growth curve. In analyzing the growth curve, it appears to be a classic sigmoidal growth curve where tumor volume increases slowly but steadily up to ~450, and then the slope of the line changes and tumor volume increases exponentially from ~500 to just below 2000 after which it plateaus at ~2000. When determining where to establish the exponential growth group, it is unclear as to why this

group was selected at the point where exponential growth is initiated (~500). Please provide a rationale for the selection of this tumor volume.

We thank the reviewer #3 for these comments, we changed the labeling for the Figure 2A. We changed the “exponential growth” for “active growth”, since it better represents the state of TRAMP-C2 cancer development as pointed out by reviewer #3.

In Fig.2B, the conclusion stated that only Bacteroidetes showed a statistical change in abundance. Please note the changes in Firmicutes abundance during exponential growth and endpoint stages. Are these changes not statistically significant? If not, how do these observations relate to those observed in human fecal material in Figure 1?

We added the statistical analysis to Figure 2B. As reviewer #3 pointed out, the changes in the relative abundance of *Firmicutes* and *Bacteroidetes* is not consistent across our mouse models and prostate cancer patients. However, we systematically observed a significant reduction in alpha diversity between all mouse models and human cohorts in relation to tumor mass. We therefore added sentences in the discussion section to clearly address the discrepancy between the different cancer cohorts and mouse models.

Page 12, lines 407-415: “Although we observed a consistent loss of alpha diversity in patients and mouse models of PCa, we did not find a conserved set of taxa systematically altered in PCa samples. Patient-to-patient variability is a common feature of human gut microbiota studies and is not unique to our study^{5,6}. As previously reported in the literature⁷, the baseline microbiota was also different between the PCa mouse models, which also possibly impaired our capacity to identify consistent changes in gut taxa as a result of PCa development. Repeated measures, paired analysis and whole-genome sequencing could help improve our capacity to identify consistent changes in gut taxa as a result of PCa development”.

The experimental approach and analysis of the data in panels 2C and 2F are limited. Using panel 2C as example, is 2C a bicompositional biplot? What samples do the left and right panels represent? Clearly the samples on the left panel show more separation where the “exponential growth” and “end point” groups cluster together, whereas the samples on the right panel show significant overlap.

We added more information for the beta-diversity analysis plots in the revised Figure 1 and 2 figure legend. The statistical analysis and p-value are also described.

The rationale for the inclusion of Pten^{-/-};Rb1^{+/+} and Pten^{-/-};Rb1^{-/-} PCa cell-derived tumors appeared to be that they were available. Please provide a rationale for including these PCa cell-derived tumors. For example, is there any correlation between the Pten^{-/-};Rb1^{+/+} and Pten^{-/-};

Rb1^{-/-} and TRAMP-C2 tumors and human PCa (e.g., genotype; histopathology; etiology, etc.), that may have modulated the impact of hFMT on tumor growth? This information would be useful at this point in the manuscript, and not just in the Discussion. In addition, the protocols on when tumor cells were injected and when treatments were given were not always clear (please see Comment 3b).

As suggested by the reviewer we provided a clear rationale for testing the *Pten* and *Rb1* double knock-out model and pinpointed the similarities between our three different prostate cancer mouse models.

Page 6, lines 198-200: “The *Pten*^{-/-}; *Rb1*^{-/-} prostate cancer cells closely resemble the genetic makeup of human metastatic prostate cancer cells and were therefore used to emulate our previous findings in patients with high PSA BCR^{18,19}”.

7. Results section, Figure 3. Please consider Comments 3d through 3f when describing the samples, data, analysis and interpretation of the data.

In Panel 3C and D4, the authors conclude that, “Interestingly, human PCa FMT2 and FMT3, corresponding to the patients with high tumor volume at radical prostatectomy, but not FMT1 corresponding to a patient with low tumor burden (Fig.3C), significantly increased TRAMP-C2 tumor burden of the recipient animals compared to the control mice at the experimental endpoint (Fig.3D).” While a comparison of the three hFMTs to the mFMT control suggests that hFMT2 and hFMT3 cluster together and are statistically significant from hFMT1, a comparison of the three hFMTs to each other would probably indicate that given the considerable overlap of the error bars, all three hFMTs stimulated TRAMP-C2 tumor growth to a similar degree,.

In the event that all three hFMTs stimulate TRAMP-C2 tumor growth to a similar degree, this might suggest that the degree of tumor burden has little effect. For consideration is that the formula used for calculating tumor burden could have impacted the analysis of these data.

Alternatively, there is a phenomenon called “super-donors” where one donor’s fecal material results in significantly more successful FMT outcomes than that of other donors. Clearly, additional samples would need to be tested to confirm any results since only 1 low tumor burden and 2 high tumor burden-derived FMTs have been tested.

We thank reviewer #3 for these comments and as suggested we performed additional experiments to provide more data for Figure 3. We added hFMTs from new donors and hFMT3 was repeated twice as a positive control.

The “super-donors” phenomenon described by the reviewer is an interesting idea. We tested the fecal samples from recipient mice for viable bacteria content using GAM broth cultures and we were surprised to observe significantly less bacteria in the fecal cultures of mice derived from high

tumour volume donors (revised Figure 3E). This suggest that the microbiota of patients with high tumour volume is not contributing to the same amount of gut microbiota colonisation compared to the autologous or low and medium tumour volume donors. This later finding is in accordance with the observed reduced alpha diversity which could translate into “poor-donors” for the microbiota of patients with high tumour volume. We reworded the manuscript to better reflect this new data and to address the comments from the reviewer.

Page 6, lines 217-236: “To test whether the gut microbiota influence PCa growth, we used fecal microbiota transplantation (FMT) in the syngeneic TRAMP-C2 mouse model of human feces collected pre-prostatectomy from 6 patients with Grade Group 2 to 5 PCa and distinct tumour volumes at surgery (Supplementary Table 3). Animals were first depleted of their microbiota using antibiotics given orally. Antibiotics effectively depleted gut microbiota as measured by total DNA extracted from fecal samples (Fig.3A) and Gifu anaerobic broth cultures from fecal samples (Fig.3B). After a 48h wash-out, mice were inoculated with the six different human fecal suspensions and a control mouse homologous tumour-free FMT that was harvested before the antibiotic regimen (n=4/FMT except for FMT3 repeated once n=8). After two series of FMT, TRAMP-C2 PCa cells were injected s.c. in both flanks of the mice and tumour mass (g) was assessed at 5 weeks. While human PCa FMT1 and FMT6 showed a trend to significantly increase TRAMP-C2 tumor mass compared to autologous mouse FMT, FMT2, 3, 4 and 5 all led to significantly bigger tumours in the recipient animals (Fig.3C). Although there was a trend for increased TRAMP-C2 volume with human FMTs from patients with increasing tumour volume, overall this was not statistically significant (Fig.3D). Noticeably, cultures derived from the FMTs of patients with high tumour volume with the lowest microbiota diversity had significantly less viable bacteria compared to the other FMTs (Fig 3E). This could suggest that the low diversity of the microbiota could impair fecal transplantation. These results however support a crosstalk by which the gut microbiota composition also influences PCa growth”.

In the Figure 3 legend, please define the acronym “J4”.

After 4 days. This was corrected in the Figure 3 legend.

8. Results section, Figures 4 and 5. In general, p values indicate that a change is statistically significant within the parameters of the experimental design; however, they do not describe what the change is. As indicated above, these data are basically descriptive, and changes (increases or decreases) are never quantified. Of note is that the y axes in figures 4C and 4D are an order of magnitude different when analyzing *Lactobacillus* 16S rRNA in fecal samples from mice with TRAMP-C2 tumors compared to fecal samples in mice without tumor, with *Pten*^{-/-}; *Rb1*^{+/+} or *Pten*^{-/-}; *Rb1*^{-/-}, indicating that *Lactobacillus* load in fecal samples from the transgenic models is considerably greater than that from the TRAMP-C2 model. In contrast, *Parabacteroidetes* load is greater (albeit more modestly) in fecal samples from the TRAMP-C2 model as compared to the transgenic mice. Therefore, while the results indicate that there are common gut microbiome

alterations, the degree of these alterations might be tumor specific.

We made changes in the manuscript to clarify the degree of differences between the two prostate cancer mouse models and their respective changes in gut microbiota. We highlighted the fact that our observations could be harder to generalize since evidence in the literature showed that a specific mouse strain could still display significant difference in their microbiota as a consequence of being provided by different vendors⁷. Page 12, lines 411-415: “As previously reported in the literature⁷, the baseline microbiota was also different between mouse models, which also possibly impaired our ability to identify consistent changes in gut taxa as a result of PCa development.”

References

- 1 Louis, P., Hold, G. L. & Flint, H. J. The gut microbiota, bacterial metabolites and colorectal cancer. *Nat Rev Microbiol* **12**, 661-672, doi:10.1038/nrmicro3344 (2014).
- 2 Irrazabal, T., Belcheva, A., Girardin, S. E., Martin, A. & Philpott, D. J. The multifaceted role of the intestinal microbiota in colon cancer. *Mol Cell* **54**, 309-320, doi:10.1016/j.molcel.2014.03.039 (2014).
- 3 Okumura, S. *et al.* Gut bacteria identified in colorectal cancer patients promote tumourigenesis via butyrate secretion. *Nat Commun* **12**, 5674, doi:10.1038/s41467-021-25965-x (2021).
- 4 Anhe, F. F. *et al.* A polyphenol-rich cranberry extract protects from diet-induced obesity, insulin resistance and intestinal inflammation in association with increased *Akkermansia* spp. population in the gut microbiota of mice. *Gut* **64**, 872-883, doi:10.1136/gutjnl-2014-307142 (2015).
- 5 Fassarella, M. *et al.* Gut microbiome stability and resilience: elucidating the response to perturbations in order to modulate gut health. *Gut* **70**, 595-605, doi:10.1136/gutjnl-2020-321747 (2021).
- 6 Costea, P. I. *et al.* Enterotypes in the landscape of gut microbial community composition. *Nat Microbiol* **3**, 8-16, doi:10.1038/s41564-017-0072-8 (2018).
- 7 Ericsson, A. C. *et al.* Effects of vendor and genetic background on the composition of the fecal microbiota of inbred mice. *PLoS One* **10**, e0116704, doi:10.1371/journal.pone.0116704 (2015).
- 8 Foster, B. A., Gingrich, J. R., Kwon, E. D., Madias, C. & Greenberg, N. M. Characterization of prostatic epithelial cell lines derived from transgenic adenocarcinoma of the mouse prostate (TRAMP) model. *Cancer Res* **57**, 3325-3330 (1997).
- 9 Haffner, M. C. *et al.* Genomic and phenotypic heterogeneity in prostate cancer. *Nat Rev Urol* **18**, 79-92, doi:10.1038/s41585-020-00400-w (2021).
- 10 Epstein, J. I., Amin, M. B., Reuter, V. E. & Humphrey, P. A. Contemporary Gleason Grading of Prostatic Carcinoma: An Update With Discussion on Practical Issues to Implement the 2014 International Society of Urological Pathology (ISUP) Consensus Conference on Gleason Grading of Prostatic Carcinoma. *Am J Surg Pathol* **41**, e1-e7, doi:10.1097/PAS.0000000000000820 (2017).
- 11 Epstein, J. I. *et al.* The 2014 International Society of Urological Pathology (ISUP) Consensus Conference on Gleason Grading of Prostatic Carcinoma: Definition of Grading Patterns and

- Proposal for a New Grading System. *Am J Surg Pathol* **40**, 244-252, doi:10.1097/PAS.0000000000000530 (2016).
- 12 Epstein, J. I. *et al.* A Contemporary Prostate Cancer Grading System: A Validated Alternative to the Gleason Score. *Eur Urol* **69**, 428-435, doi:10.1016/j.eururo.2015.06.046 (2016).
- 13 Hennes, D. *et al.* The modified International Society of Urological Pathology system improves concordance between biopsy and prostatectomy tumour grade. *BJU Int* **128 Suppl 3**, 45-51, doi:10.1111/bju.15556 (2021).
- 14 De Nunzio, C. *et al.* The new Epstein gleason score classification significantly reduces upgrading in prostate cancer patients. *Eur J Surg Oncol* **44**, 835-839, doi:10.1016/j.ejso.2017.12.003 (2018).
- 15 Corcoran, N. M. *et al.* Upgrade in Gleason score between prostate biopsies and pathology following radical prostatectomy significantly impacts upon the risk of biochemical recurrence. *BJU Int* **108**, E202-210, doi:10.1111/j.1464-410X.2011.10119.x (2011).
- 16 Sowalsky, A. G. *et al.* Gleason Score 7 Prostate Cancers Emerge through Branched Evolution of Clonal Gleason Pattern 3 and 4. *Clin Cancer Res* **23**, 3823-3833, doi:10.1158/1078-0432.CCR-16-2414 (2017).
- 17 Cooper, C. S. *et al.* Analysis of the genetic phylogeny of multifocal prostate cancer identifies multiple independent clonal expansions in neoplastic and morphologically normal prostate tissue. *Nat Genet* **47**, 367-372, doi:10.1038/ng.3221 (2015).
- 18 Nyquist, M. D. *et al.* Combined TP53 and RB1 Loss Promotes Prostate Cancer Resistance to a Spectrum of Therapeutics and Confers Vulnerability to Replication Stress. *Cell Rep* **31**, 107669, doi:10.1016/j.celrep.2020.107669 (2020).
- 19 Hamid, A. A. *et al.* Compound Genomic Alterations of TP53, PTEN, and RB1 Tumor Suppressors in Localized and Metastatic Prostate Cancer. *Eur Urol* **76**, 89-97, doi:10.1016/j.eururo.2018.11.045 (2019).

REVIEWER COMMENTS

Reviewer #1 (Remarks to the Author):

I have read the revised manuscript and found that the authors have adequately addressed all my concerns. Therefore, I support the publication of the revised manuscript in Nature Communications.

Reviewer #2 (Remarks to the Author):

Thank you to the authors for their through response, which sufficiently addresses all the issues that I raised. I have no further questions and will leave the final decision to the editor(s).

Reviewer #3 (Remarks to the Author):

General comments:

1. The authors have thoughtfully addressed all of the reviewer's comments. The findings of this study are also significantly strengthened by the increase in patient number per cohort and by providing more detailed descriptions of the patient cohorts, the experimental design, data analyses, etc.

2. The authors indicate that the Pten^{-/-}; Rb1^{-/-} mouse model is "a model that replicate patients metastatic PCa where loss of Rb protein is common", and that this observation is verified in the literature. While not critical for this manuscript, genotyping the human tumors would have provided direct proof that the metastatic human tumors analyzed in this study did not express Pten and/or Rb protein and were therefore comparable with the mouse-derived Pten^{-/-}; Rb1^{-/-} prostate tumor models.

3. The authors have revised a considerable number of statements in an attempt to make them more analytical. However, overall, the revisions simply restate the initial statements. Using one revision as an example, the statement “We also noted a significant reduction of Bacteroidetes relative abundance in BCR high PSA was significantly lower compared to no BCR patients (Fig.1F, green bars).” was revised to, “We also noted that Bacteroidetes relative abundance in BCR high PSA was significantly lower compared to no BCR patients (Fig.1F, green bars, $p=1.5e-6$, Welch’s t test)”.

While a p value is added to the revised version, the statement fundamentally remains the same, as “significant reduction” = “significantly lower”. Nearly all the other statements describing data were revised in this manner.

Data should be reported as accurately as possible, especially when significant differences are observed. Therefore, the example statement above could be rewritten, “The relative abundance of Bacteroidetes in the BCR/high PSA group decreased significantly by 80% ($p=1.5e-6$) compared to that observed in the no BCR/PSA- group (Fig.1F)”. This statement provides a numerical value for comparison and clearly highlights the level (and impact) of the decrease in Bacteroidetes abundance in one group over the other.

Nature Communications manuscript NCOMMS-22-05203A

Reviewer #1 (Remarks to the Author):

I have read the revised manuscript and found that the authors have adequately addressed all my concerns. Therefore, I support the publication of the revised manuscript in Nature Communications.

Reply: We thank reviewer #1 for reviewing our manuscript and supporting the publication.

Reviewer #2 (Remarks to the Author):

Thank you to the authors for their through response, which sufficiently addresses all the issues that I raised. I have no further questions and will leave the final decision to the editor(s).

Reply: We thank reviewer #2 for his comment and for reviewing our manuscript.

Reviewer #3

1. The authors have thoughtfully addressed all of the reviewer's comments. The findings of this study are also significantly strengthened by the increase in patient number per cohort and by providing more detailed descriptions of the patient cohorts, the experimental design, data analyses, etc.

2. The authors indicate that the *Pten*^{-/-}; *Rb1*^{-/-} mouse model is “a model that replicate patients metastatic PCa where loss of Rb protein is common”, and that this observation is verified in the literature. While not critical for this manuscript, genotyping the human tumors would have provided direct proof that the metastatic human tumors analyzed in this study did not express *Pten* and/or *Rb* protein and were therefore comparable with the mouse-derived *Pten*^{-/-}; *Rb1*^{-/-} prostate tumor models.

3. The authors have revised a considerable number of statements in an attempt to make them more analytical. However, overall, the revisions simply restate the initial statements. Using one revision as an example, the statement “We also noted a significant reduction of Bacteroidetes relative abundance in BCR high PSA was significantly lower compared to no BCR patients (Fig.1F, green bars).” was revised to, “We also noted that Bacteroidetes relative abundance in BCR high PSA was significantly lower compared to no BCR patients (Fig.1F, green bars, $p=1.5e-6$, Welch's t test)”.

While a p value is added to the revised version, the statement fundamentally remains the same, as “significant reduction” = “significantly lower”. Nearly all the other statements describing data were revised in this manner.

Data should be reported as accurately as possible, especially when significant differences are observed. Therefore, the example statement above could be rewritten, “The relative abundance of Bacteroidetes in the BCR/high PSA group decreased significantly by 80% ($p=1.5e-6$) compared to that observed in the no BCR/PSA- group (Fig.1F)”. This statement provides a numerical value for comparison and clearly highlights the level (and impact) of the decrease in Bacteroidetes abundance in one group over the other.

We thank reviewer #3 for his comments. We took good note of his comments on *Pten* and *Rb1* genotyping. We have addressed the remaining issues regarding data reporting accuracy, these changes are highlighted in blue in the revised manuscript.